# ROBUST PREDICTION UNDER MISSINGNESS SHIFTS

## ABSTRACT

Prediction becomes more challenging with missing covariates. What method is chosen to handle missingness can greatly affect how models perform. In many real-world problems, the best prediction performance is achieved by models that can leverage the informative nature of a value being missing. Yet, the reasons why a covariate goes missing can change once a model is deployed in practice. If such a missingness shift occurs, the conditional probability of a value being missing differs in the target data. Prediction performance in the source data may no longer be a good selection criterion, and approaches that do not rely on informative missingness may be preferable. However, we show that the Bayes predictor remains unchanged by ignorable shifts for which the probability of missingness only depends on observed data. Any consistent estimator of the Bayes predictor may therefore result in robust prediction under those conditions, although we show empirically that different methods appear robust to different types of shifts. If the missingness shift is non-ignorable, the Bayes predictor may change due to the shift. While neither approach recovers the Bayes predictor in this case, we found empirically that disregarding missingness was most beneficial when it was highly informative.

## 1 INTRODUCTION

In many applied domains, such as healthcare, predictive models are trained on data that contains missing values. For example, magnetic resonance imaging (MRI) is expensive and only conducted when clinically indicated. As a result, not all patients will have an MRI recorded, and patients with and without MRI results may differ systematically (Rubin, 1987). How missing covariates are handled can greatly impact prediction performance (Perez-Lebel et al., 2022). Most approaches follow a two-stage process: find a (good) imputation, and then fit a standard prediction model to the completed data (Wood et al., 2015; Le Morvan et al., 2021). Others bypass imputation and learn separate submodels for each missingness pattern (Fletcher Mercaldo & Blume, 2020) or optimally embed the missing data for prediction (Le Morvan et al., 2020).

Which approach is ultimately chosen usually depends on the prediction performance in held-out test data. Performance-based model selection tends to favour models that can also exploit information encoded in the missingness (Perez-Lebel et al., 2022; Rockenschaub et al., 2023; Sisk et al., 2023). Exclusively focusing on prediction performance ignores a key challenge, though: the mechanisms that govern which covariates are observed and which are missing may — and often do — change once a model is deployed in the real world (Sperrin et al., 2020). For instance, reduced costs can influence how often clinical tests are performed, with the recent increase in MRIs as a prime example (Smith-Bindman et al., 2019). Clinical guidelines may similarly evolve over time. In either case, missingness changes and no longer carries the same meaning, potentially invalidating the learned relationships. Even the introduction of the model itself can shift how and when data are collected (Sperrin et al., 2020).

This immediately raises two critical questions which form the core of this paper: If the mechanisms of missingness change during deployment, can methods that leverage informative missingness still maintain their edge? Is there an advantage in using methods that might be less performant but do not rely on informative missingness? To answer these questions, this paper discusses the theoretical conditions under which predictions remain reliable when missingness changes, and empirically evaluates the robustness of several common prediction methods under those conditions. In doing so, we make the following contributions:

- We formalise the effect of missingness shifts on the optimal predictor from a missing data perspective, showing that even estimators that utilise informative missingness remain valid if missingness is ignorable in both the source and target environment.

- We demonstrate that although all estimators may theoretically achieve robust predictions under ignorable missingness shifts, they do not always do so reliably.

- We introduce NeuMISE, a novel neural architecture that provides competitive prediction performance while remaining comparatively robust in the presence of ignorable shifts.

## 2 RELATED WORK

**Learning with missing data** Rubin (1976) established conditions for valid inference from partially missing data (see Appendix C for a detailed summary). Without information on the exact mechanisms that influence missingness, they showed that unbiased estimation of the full data distribution is possible if data is *Missing (Completely) At Random (M(C)AR)* — i.e., if missingness only depends on observed values. Missingness of this types is said to be *ignorable* (Rubin, 1976). If missingness depends on unobserved quantities, however, data is considered *Missing Not At Random (MNAR)*. Additional information not deducible from the data alone is necessary for valid inference (Mohan & Pearl, 2021). Since this is hard to come by in practice, most approaches for dealing with missing data assume data are *M(C)AR*, which may often be implausible (Le Morvan et al., 2021).

**Impute-then-regress** Since most off-the-shelf prediction models require fully-observed features, missing values are frequently imputed prior to model training. Methods for imputation range from conditional mean imputation to full estimation of the posterior probability of the missing values. The latter is motivated by Rubin (1987), which promises unbiased estimation of the complete data model[1] if done correctly. One of the best known imputation algorithms is Imputation by Chained Equations (ICE), which iteratively trains one conditional imputation model per variable (White et al., 2011). Increasingly, machine learning is used to generate imputations, either as part of ICE (Van Buuren, 2018) or directly through the use of generative models such as (variational) autoencoders (Mattei & Frellsen, 2019) or generative adversarial networks (Yoon et al., 2018).

**Prediction without imputation** Rosenbaum & Rubin (1984) first considered fitting a separate submodel per missingness pattern, thus bypassing the need for imputation and allowing the model to utilise information encoded in the missingness pattern. This idea was revisited by Josse et al. (2020), who showed that pattern submodels minimise the empirical loss. Le Morvan et al. (2021) further proved that any non-probabilistic impute-then-regress approach asymptotically estimates a pattern submodel, provided that a suitably flexible predictor is used. However, the difficulty of estimating a model for all possible missingness patterns — which grow exponentially with the number of covariates — was already noted by Rosenbaum & Rubin (1984). The potential sparsity of data may be avoided by considering only common patterns (Fletcher Mercaldo & Blume, 2020) or by allowing patterns to share information (Le Morvan et al., 2020).

**Distribution shift** Distribution shifts are an active area of research, with a strong focus on covariate or label shifts (Cai et al., 2023). Changes in missingness have been relatively understudied and mostly limited to illustrative examples (Groenwold, 2020) or empirical investigations (Jeanselme et al., 2022). Only very recently, Zhou et al. (2023) formally introduced the notion of missingness shifts in the context of domain adaptation, showing that they reduce to covariate shifts if the missingness is ignorable. Here, we expand on this result from a missing data perspective.

## 3 PROBLEM DEFINITION

**Covariates** Following Le Morvan et al. (2021), we consider $N$ i.i.d. realisations of the random vector $(X, M, Y) \in \mathbb{R}^d \times \{0, 1\}^d \times \mathbb{R}$, where $X$ are the complete (counterfactual) covariates, $M$ are the missingness indicators, and $Y$ is the outcome. For a single $(x, m, y)$ realisation, $m^{(j)} = 1$ indicates that $x^{(j)}$ is missing and $m^{(j)} = 0$ means that it is observed. The complete covariates $X$ are generally not available in practice. Instead, we only have access to the partially-observed covariates $\tilde{X} \in (\mathbb{R} \cup \{\texttt{NA}\})^d$ denoted as

---

[1]That is, the model that minimises the prediction loss had we had completely observed data.

$$\tilde{X}^{(j)} = \begin{cases} X^{(j)} & \text{if } M^{(j)} = 0, \\ \text{NA} & \text{if } M^{(j)} = 1, \end{cases} \tag{1}$$

where $j \in \{1, ..., d\}$ indicates the covariate and NA represents a missing value. We further denote the indices corresponding to observed features by $obs(M) \subseteq \{1, ..., d\}$ and missing features by $mis(M) = \{1, ..., d\} \setminus obs(M)$. Hence, the observed features are denoted as $X_{obs(M)}$ and missing features as $X_{mis(M)}$. For readability, we use $X_{obs}$ and $X_{mis}$ in the remainder.

**Outcome** Without loss of generality, we assume that the outcome $Y$ is generated according to

$$Y = f^\star(X) + \epsilon, \quad \text{with } \mathbb{E}[\epsilon \mid X_{obs}, M] = 0 \text{ and } \mathbb{E}[Y^2] < \infty. \tag{2}$$

An additive noise was chosen to simplify $Y$ into a deterministic real-valued function $f^\star : \mathbb{R}^d \to \mathbb{R}$ that only depends on the complete data $X$, and a random term $\epsilon \in \mathbb{R}$. However, the results in this work also hold for binary outcomes and more general noise terms (Appendix A.2). Unless stated otherwise, we further assume that the outcome does not affect the probability of a value being missing, i.e., $p(M \mid X, Y) = p(M \mid X)$. We will relax this assumption later.

**Objective** If missing data may be present at the time of prediction, we require a model that can make optimal predictions given only partially-observed data. This is called the Bayes predictor (Le Morvan et al., 2021) and is given by

$$\tilde{f}^\star(\tilde{X}) = \mathbb{E}[Y \mid X_{obs}, M] = \mathbb{E}[f^\star(X) \mid X_{obs}, M], \tag{3}$$

where the second equality derives from our simplifying assumptions in Equation (2). The Bayes predictor $\tilde{f}^\star$ is related to — but generally differs from — $f^\star$, the true complete data model. Equation (3) shows that $\tilde{f}^\star$ may rely on information encoded by the missingness indicator $M$. In fact, there are up to $2^d$ different Bayes predictors $\tilde{f}_m^\star \in \mathcal{F}$, with one predictor $\tilde{f}_m^\star$ per missingness pattern $m$. To highlight the relationship between the two functions, each pattern-specific Bayes predictor $\tilde{f}_m^\star$ may be written as an integral of $f^\star$ over all possible values of $X_{mis}$ given observed values $X_{obs}$ and pattern $m$:

$$\tilde{f}_m^\star(\tilde{X}) = \int_{X_{mis}} f^\star(X_{obs}, X_{mis}) p(X_{mis} \mid X_{obs}, M = m) dX_{mis}. \tag{4}$$

**Missingness shifts** It is usually assumed that the probability of missingness $p(M \mid \cdot)$ remains unchanged when moving from the training (source) to the test (target) environment (Sperrin et al., 2020). However, as we motivated in the introduction, changes in missingness are likely the norm.

**Definition 1 (Missingness shift)** *Given a source and target environment with probabilities $p(\cdot)$ and $q(\cdot)$, we say there is a shift in the missingness if $\exists m \in \{0, 1\}^d : p(M = m \mid X) \neq q(M = m \mid X)$.*

## 4 ROBUST PREDICTION UNDER MISSINGNESS SHIFT

The optimal prediction with missing data is given by the Bayes predictor in Equation (3). Once we allow for shifts in missingness, we may therefore ask under which conditions the Bayes predictor of the source environment remains optimal for the target environment. Assuming no other shifts with $p(Y, X) = q(Y, X)$, let $\tilde{f}_m^\star$ and $\tilde{g}_m^\star$ be the Bayes predictor(s) for the source and target environment respectively. Zhou et al. (2023) first showed that $\forall m \in \{0, 1\}^d : \tilde{f}_m^\star = \tilde{g}_m^\star$ if missingness only depends on the observed covariates $X_{obs}$. We restate this result in Theorem 1 and introduce the notion of ignorable missingness shifts:

**Definition 2 (Ignorable missingness shift)** *A missingness mechanism $p(M \mid \cdot)$ is considered ignorable if $\forall m \in \{0, 1\}^d : p(M = m \mid \cdot) = p(M = m \mid X_{obs})$, that is, if given the observed data, the missingness does not depend on unobserved covariates. By extension, a missingness shift is ignorable if both the source missingness $p(M \mid \cdot)$ and target missingess $q(M \mid \cdot)$ are ignorable.*

**Theorem 1 (Equivalence of Bayes predictors under missingness shift)** *Assume that the data is generated according to Equation (2). Then, if there is a missingness shift, $\forall m \in \{0,1\}^d :$ $\tilde{f}^\star{}_m = \tilde{g}^\star{}_m$ holds in general only if the missingness shift is ignorable.*

A slightly adapted proof starting from Equation (4) can be found in Appendix A.1. Theorem 1 states that the Bayes predictors remain unchanged across environments even if the missingness mechanisms vary, but only if missingness is ignorable in both environments. If missingness is not ignorable in either the source and/or the target environment, the Bayes predictors may differ and robust prediction is no longer guaranteed.

### 4.1 Importance of unbiased estimation of $f^\star$ for robust prediction

Many approaches for dealing with missing data — both traditional (Rubin, 1976; 1987) and more recent (Ipsen et al., 2022; Jarrett et al., 2022) — assume ignorable missingness (Le Morvan et al., 2021). Under these conditions, one may obtain an unbiased estimate of the complete data model $f^\star$ (Rubin, 1976). This is obviously important if the parameters of $f^\star$ are themselves of scientific interest, but unbiased estimation has also been portrayed as desirable for robust prediction (Steyerberg & Vergouwe, 2014; Wynants et al., 2020; Ipsen et al., 2022). Theorem 1 calls this into question. It suggests that unbiased estimation of $f^\star(X)$, for example through the use of multiple imputation (Van Buuren, 2018), may not be necessary for robust prediction under ignorable missingness. All that is required is precise estimation of the Bayes predictors $\tilde{f}_m$ in the source environment. While this is still a daunting task due to the combinatorial difficulties of estimating $|\mathcal{F}| = 2^d$, potentially very different Bayes predictors, Le Morvan et al. (2021) showed that many common predictors including simple methods like mean imputation arrive at the Bayes predictor, at least in the limit. There is therefore no a priori reason to prefer unbiased estimation. On the other hand, if missingness is not ignorable to begin with, then any estimators that assumes ignorability will not be unbiased either. In this case, only explicit specification of the missingness mechanism may lead to robust prediction.

### 4.2 Complete observations as a special case

The traditional goal of statistical inference with missing data is the unbiased estimation of the complete data model $f^\star(X)$. This can be seen as a special case of prediction under missingness shift. Moving from a source environment with missingness to fully observed data constitutes a missingness shift with $q(M \neq 0 \mid X) = 0$. Here, we are only interested in a single Bayes predictor, namely $\tilde{g}^\star{}_{[0,\ldots,0]}$, and classic statistical results tell us that $\tilde{g}^\star{}_{[0,\ldots,0]}$ can be estimated if the missingness mechanism in the source environment is ignorable. Theorem 1 extends this to all $\tilde{g}^\star{}_m \in \mathcal{G}$ and can be seen as a generalisation of the well-known result of unbiased estimation under ignorability.

## 5 Learning under ignorable missingness shifts

Theorem 1 implies that for ignorable missingness shifts, any consistent estimator of the Bayes predictors should lead to robust prediction in the limit. However, existing methods differ in how they estimate Equation (3), which may affect their sensitivity to changes in the missingness mechanism.

**Impute-then-regress** procedures follow the factorisation in Equation (4), imputing with $p(X_{mis} \mid X_{obs}, M)$ [2] and using the filled-in data to learn $f^\star(X_{obs}, X_{mis})$. The success of this approach depends on our ability to estimate the imputation model $p(X_{mis} \mid \cdot)$ well (Le Morvan et al., 2021). If we had perfect knowledge of the imputation model, we could repeatedly draw from it to approximate the integral in Equation (4) as closely as we desire. Factors like high rates of missingness, however, may affect the quality of the imputation model and limit our ability to approximate Equation (4).

**Prediction without imputation** directly estimates Equation (3). (Le Morvan et al., 2020) recently proposed a neural network-based approach called NeuMiss to solve the combinatorial issues involved. NeuMiss embeds $\tilde{X}$ using the inverted observed-data covariance $\Sigma_{obs}^{-1}$, which is approximated through a recursive neural network performing a truncated Neumann series. To do so, $\tilde{X}$ is first centred and zero-filled. We then obtain an initial embedding $\tilde{X}_0 = (V\tilde{X}) \odot \bar{m} + \tilde{X}$, where $V$ is

---

[2]Or $\mathbb{E}[X_{mis} \mid X_{obs}, M]$ in the case of conditional mean imputation.

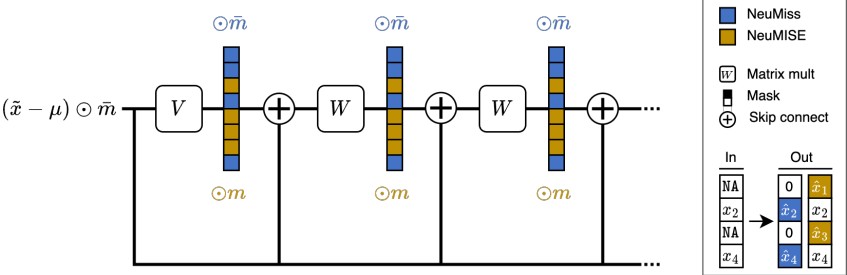

Figure 1: Proposed architecture of NeuMISE compared to the standard NeuMiss architecture.

a weight matrix, $\bar{m} = 1 - m$ is the inversed missingness indicator, and $\odot$ is an element-wise multiplication (see Figure 1 and Appendix E for details). At each subsequent iteration $i$, $\tilde{X}_i$ is replaced by $W\tilde{X}_{(i-1)} \odot \bar{m} + \tilde{X}$. After $n$ iterations, prediction is performed on $\tilde{X}_n$ by a simple fully-connected layer, which is learned together with the embedding in an end-to-end fashion. Crucially, through $\Sigma_{obs}^{-1}$, information is shared between missing patterns $m \in \{0,1\}^d$. Le Morvan et al. (2021) showed that this approach performs well in i.i.d data. Directly learning $\tilde{f}_m^\star$ avoids the need to estimate an imputation model $p(X_{mis} \mid \cdot)$, making it suitable for situations in which correct imputations may be hard to learn. However, prediction without imputation may struggle to extrapolate to previously unseen missingness patterns $m$ unless it has a reliable way of sharing information between patterns.

**Other methods** for learning with missing data exist. In particular, Inverse Probability Weighting (IPW) is common in statistical inference with missingness Seaman & White (2013). IPW is less commonly used for prediction, though, and we do not consider it further in this work. However, we note that in the absence of information about the target environment, the domain adaptation proposed in Zhou et al. (2023) reduces to IPW (Appendix A.5).

### 5.1 ROBUST END-TO-END LEARNING: NEUMISE

To see why extrapolation to previously unseen patterns might pose a problem for NeuMiss in particular, consider a sample with fully observed covariates, i.e., $X_{obs} = X$. Intuitively, we could simply forward $X$ to a fully-connected layer that performs the prediction. Yet, NeuMiss first embeds this as $(\Sigma)^{-1}X$ (Le Morvan et al., 2021). If few or no samples with complete covariates were observed during training, it is unlikely that this embedding is meaningful. We further observe that this is a general behaviour of NeuMiss: as the number of observed covariates increases, the number of dimensions $|obs(M)|$ used for embedding also increases, embedding fewer missing covariates in more dimensions. We hypothesise that this may hurt NeuMiss' ability to extrapolate.

We propose a simple change to the architecture of NeuMiss that addresses this issue. Rather than embedding missing information in $(\Sigma_{obs})^{-1}$, we propose an imputation-like embedding that targets $\Sigma_{mis,obs}(\Sigma_{obs})^{-1}$ instead. That is, $\tilde{X}$ is embedded in the missing dimensions $|mis(M)|$ rather than superimposed on the observed covariates. At each iteration, we update the embedding with $\tilde{X}_i = W\tilde{X}_{(i-1)} \odot m + \tilde{X} \odot \bar{m}$. For fully observed covariates NeuMISE simply forwards the unchanged vector $X$. Notably, this only differs from NeuMiss in the masking used in the recursive layer (Figure 1). This ostensibly simple changes leads to markedly different non-linearities within the network layer, similar to iterative imputation in ICE. Both current embedding values and observed covariates are used to update the embedding at the next step. The only other change we made is the use of batch normalisation instead of substracting $\mu$, as it empirically improved stability when missingness increased. In acknowledgment of the principles from NeuMiss and ICE that influenced our method, we termed it Neural Multivariate Imputation via Simultaneous Equations (NeuMISE).

## 6 THE ROLE OF Y

So far, we have only considered missingness governed by the covariates $X$. In practice, missingness may also depend on the outcome $Y$. For example, disease severity may affect a patient's likelihood of receiving an MRI. We write $p(M \mid X, Y)$ if the missingness depends on the outcome.

### 6.1 UNBIASED ESTIMATION INTRODUCES SHIFTS IN Y-DEPENDENT MISSINGNESS

The Bayes predictor in Equation (3) does not make any assumptions about the missingness mechanism. It remains valid even in the case of Y-dependent missingness. Furthermore, in the absence of a missingness shift, we trivially have $\forall m \in \{0,1\}^d : \tilde{f}_m^\star = \tilde{g}_m^\star$. Directly estimating $\tilde{f}_m^\star(\tilde{X})$ therefore leads to robust prediction in the target environment. Difficulties only arise if we aim for unbiased estimation of $f^\star$, for example by multiple imputing with $p(X_{mis} \mid X_{obs}, Y)$ (Wood et al., 2015). By conditioning on $Y$, the missingness in the source environment becomes ignorable and unbiased estimation of $f^\star$ is indeed possible (White et al., 2011). However, $Y$ crucially cannot be available in the target environment, or there would be no point in predicting. Missingness in the target environment will therefore always be non-ignorable, introducing an artificial missing shift caused solely by our choice of analysis, even if the underlying missingness mechanisms remained the same.

### 6.2 THE EFFECT OF SHIFTS IN Y-DEPENDENCE

The above only holds as long as there is no actual underlying missingness shift between environments. If the probability $q(M \mid X, Y)$ does change *while still depending on* $Y$, however, we inevitably have a shift with non-ignorable missingness in at least one environment (the target). By Theorem 1, the Bayes predictor therefore is no longer guaranteed to remain unchanged.

**Corollary 1 (Non-ignorability of Y-dependent missingness shifts)** *If* $q(M \mid X, Y) \neq q(M \mid X)$ *and* $q(M \mid X, Y) \neq p(M \mid X, Y)$, *then the missingness shift cannot be ignorable.*

It is worth noting that Corollary 1 requires $Y$-dependence in the target environment. If $Y$ only influences missingness in the source environment, shifts may still be ignorable. This is for example the case when we train on *MAR* data but assume no missingness in the target environment, i.e., $p(M \mid X_{obs}, Y)$ and $q(M \neq 0) = 0$. Such fully observed data in the target environment is a common case in the development of clinical scores (see for example Collins & Altman (2010; 2012); Zippl et al. (2022)). Here, adjusting for $Y$ in the imputation will provide an unbiased estimate of $f^\star$, which happens to be the Bayes predictor for the target environment. Similar conclusions should hold for the more general case of $q(M \mid X_{obs})$, although it is not immediately clear how it may be derived from $p(M \mid X_{obs}, Y)$ in practice. While we leave the investigate of this case to future work, we outline a possible solution in Appendix A.4 for illustration .

## 7 EXPERIMENTS

### 7.1 DATA

**Simulation** We followed Le Morvan et al. (2021) and simulated $N = 100,000$ observations of $d = 50$ covariates from a multivariate normal distribution $X \sim N(\mu, \Sigma)$ with $\mu \sim N(0, 1)$. The covariance was defined as $\Sigma = BB^T$, where $B \in \mathbb{R}^{d \times \lceil \lambda d \rceil}$ and $B_{i,j} \sim N(0, 1)$. $\lambda$ governed the degree of correlation between covariates, with lower $\lambda$ implying higher correlation. We used $\lambda = 0.7$ and $\lambda = 0.3$ to simulate high and low correlation settings. Using the covariates, we generated a continuous outcome $Y = h(X) + \epsilon$, where $h$ is a non-linear function of $X$ and $\epsilon$ is noise. For details on how covariates and outcomes were parameterised, please refer to Appendix D.

**Linked Birth and Infant Death Data (LBIDD)** To test our findings on real-world data, we used covariate data from the 1995 infant mortality statistics of the US National Center for Health Statistics (MacDorman & Atkinson, 1998). After extracting 50 complete covariates $X$ from the dataset (binary and continuous), we applied the procedures described above to simulate an artificial outcome $Y$.

### 7.2 MISSINGNESS SCENARIOS

We considered the following missingness mechanisms for our experiments: for **MCAR**, each covariate was randomly and independently masked with constant probability; for **non-monotone MAR**, a subset of 30% of the covariates were MCAR and the remaining 70% of covariates were masked with a probability that depends on the remaining observed values through a logistic function; for **MNAR**, covariates were set to missing with a probability that depended on their own value via a Gaussian

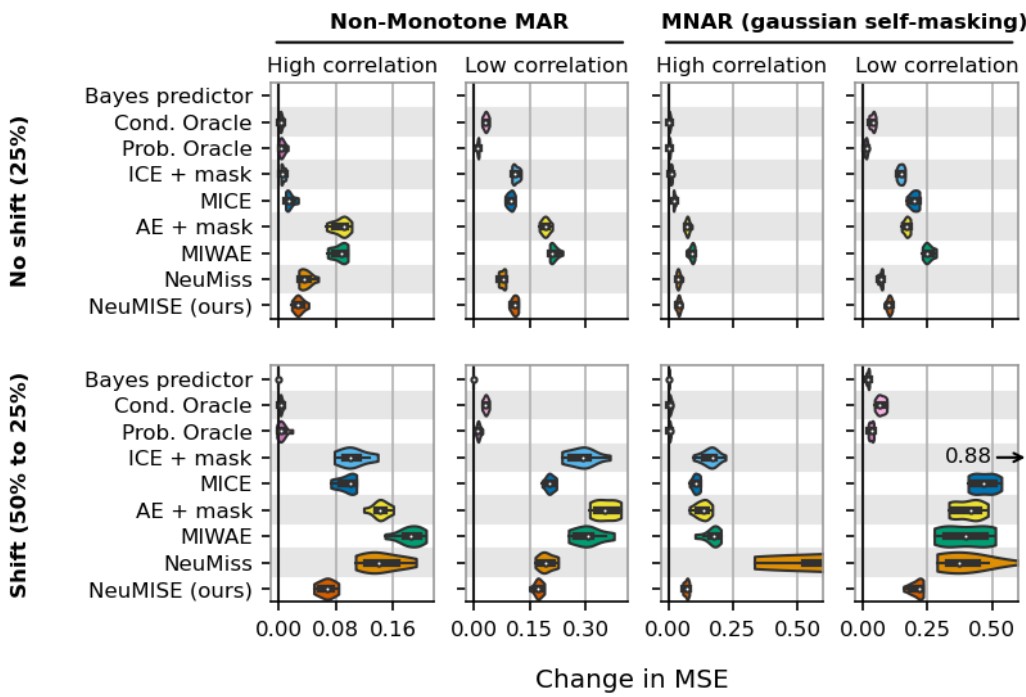

Figure 2: Change in MSE in the target environment with 25% missingness relative to the analytical Bayes predictor. We evaluated estimators trained in the target environment (no shift) and estimators trained in another source environment (shift from 50% to 25%) across 10 random reruns.

function. In addition, we considered a setting **MAR-Y** in which the missingness of covariates depended only on Y through a logistic function. Across all missingness mechanisms, we trained in a source environment with 50% missingness (original data) and evaluated in an independent target environment with 25% missingness (shifted data) or 0% missingness (complete data). Performance of each estimator in the target environment was compared to its i.i.d. equivalent trained in the target environment (no shift). For more details on all scenarios, please refer to Appendix D.

## 7.3 BASELINES

**Bayes predictor and oracles** Where the scenario allowed for analytical derivation of the Bayes predictor, we included it as the ground truth in our experiments. This is the case for MCAR, non-monotone MAR, and MNAR missingness in the simulated data (Le Morvan et al., 2021). We additionally included oracle estimators that had access to the true complete data predictor $f^\star$ and chained it with imputations from the true conditional expectation $\mathbb{E}[X_{mis} \mid X_{obs}, M]$ (**Cond. Oracle**) or conditional probability distribution $p(X_{mis} \mid X_{obs}, M)$ of the missing data (**Prob. Oracle**).

**Imputers** We considered several learnable imputers. **ICE** uses linear models to iteratively impute each variable with the conditional expectation given the values of all other variables (Van Buuren, 2018). Multiple ICE (**MICE**) relies on the same iterative process but approximates the integral in Equation (4) by drawing multiple possible values from the conditional probability distribution given by the linear model. For the MAR-Y setting, we also fitted variants that include Y for unbiased imputation during training but omit it during evaluation (**ICE-Y** and **MICE-Y**), as recommended by Wood et al. (2015). The missing data importance-weighted autoencoder (**MIWAE**) uses a variational autoencoder to maximise a lower bound of the observed-data likelihood (Mattei & Frellsen, 2019). We used this autoencoder to obtain multiple draws from the conditional probability distribution. Alternatively, we used a non-probabilistic variant that derives the conditional expectation of the missing data instead (**AE**). Finally, **NeuMiss** employs a neural network to approximate the inverse covariance matrix of the observed values through a Neumann series (Le Morvan et al., 2020). Imputers using conditional expectation imputed each missing value exactly once. Imputers that draw

from the conditional distribution drew $n_{imp} = 5$ samples per missing value, resulting in 5 multiply imputed datasets. All imputers were followed by a standard feed-forward network. For ICE and AE variants, this network was learned *separately* from the imputations. That is, no information from the outcome was fed back to the imputer (impute-then-regress). NeuMiss and NeuMISE were trained end-to-end, with gradients from the predictions also influencing the representation of missing values.

**Training and evaluation** All models were trained on $100,000$ samples and tested on an independent set of $10,000$ samples. An additional $10,000$ samples were used as validation set. Performance was evaluated using the mean squared error (MSE). Optimal hyperparameters were chosen via grid search across 10 randomly initialised repetitions. Hyperparameters included network depth and width, learning rate, and weight decay (Appendix F). All models were trained with a batch size of $100$ for a maximum of $1,000$ epochs using an Adam optimiser. Training was stopped early if the validation performance did not improve for 12 epochs. We employed a learning rate schedule, multiplying by $0.2$ when performance did not improve for 10 epochs.

## 7.4 RESULTS

**Performance in the absence of shifts** In the source environment, all estimators achieved performances close to the optimal Bayes predictor for both MAR and MNAR settings (Figures 2 and 5). Low correlations were generally harder to predict. Except for MNAR data with high missingness and low correlation (Figure 7), there was no notable difference between theoretically unbiased variants (MICE and MIWAE) and those set up to exploit missingness. In line with (Le Morvan et al., 2021), this suggests that a good approximation of the Bayes predictor does not rely on the unbiased estimation. Single imputation with ICE was quite competitive, outperforming other estimators especially under high correlation. This may be expected, as the multivariate normal covariates are a good fit for ICE. NeuMiss and NeuMISE on the other hand grew increasingly competitive with higher amounts of missingness, outperforming all models except Oracles (Figure 7).

**Equivalence of the Bayes predictor under shift** As predicted by Theorem 1, the Bayes predictor remained consistent under ignorable missingness shifts (Figures 2 and 5). The Bayes predictor was no longer stable in the MNAR setting, although increases in MSE were minor — or even negligible in the case of high correlation. Since in both cases we have access to the Bayes predictor in closed form, we can also verify this result analytically (see Appendix E). Oracle estimators with access to the true conditional expectation or distribution of the missing covariates had increased MSE for any type of shift, but remained reasonably robust throughout the considered settings.

**Robustness of estimators to shifts** The effects of shifts were more pronounced for learned estimators (Figure 2). NeuMISE remained closest to the Bayes predictor, outperforming both unbiased estimation via MICE as well as end-to-end learning via NeuMiss. ICE+mask, which had access to the missingness pattern, was generally a little less robust than MICE but had particularly bad performance in the case of MNAR with low correlation. Notably, robustness of estimators was affected by the direction of the shift. When shifting from 25% to 50% missingness, the performance of ICE+mask and MICE *improved* compared to direct training on 50% missingness (Figure 7). This may be because lower missingness allowed them to learn a better imputation model. NeuMISE was less robust to increasing missingness, which may reflect the fact that — contrary to NeuMiss – NeuMISE's embedding dimensions increase together with the missingness.

**Unbiased estimation of the complete data predictor** When missingness only depends on the covariates $X$, all estimators are in principle able to recover the complete data predictor (see White & Carlin (2010) and Appendix C). Among the considered estimators, NeuMISE consistently achieved the best performance in complete data across missingness mechanisms (Figure 6), outperforming even unbiased estimators. This may be due to an inability of the imputation models to learn $p(X_{mis} \mid X_{obs})$ under such high levels of missingnes. NeuMiss' drop in performance was especially pronounced, repeatedly performing worse than a simple mean model (MSE $\approx 1.28$).

**Effects of Y-dependent missingness** When missingness depended on Y, ICE+mask and Neumiss were able to exploit this fact and performed even *better* than the true complete data predictor (Figure 3). Their performance in the shifted data, however, was often worse than a mean model. NeuMISE stayed close to the complete data performance in both high and low correlation settings and had more robust — but still bad — performance in the shifted data. MIWAE and MICE+Y performed worst in the absence of a shift but retained that performance when a shift occurred.

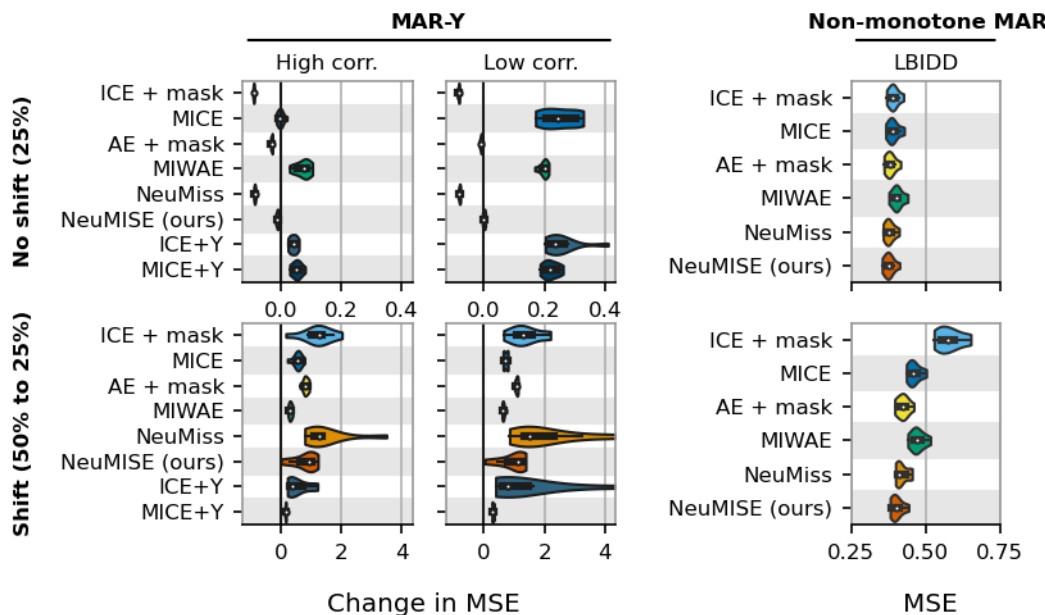

Figure 3: Change in MSE in simulated data compared to the complete data predictor. Due to dependence on $Y$, performance may be *better* than on complete data.

Figure 4: MSE in the semi-simulated LBIDD data without access to ground truth covariate data.

**Performance in real-world data** Estimators performed comparably in the semi-simulated LBIDD data, with NeuMiss and NeuMISE slightly outperforming the other estimators in shifted and un-shifted data (Figure 4). Unlike in the simulated data, ICE+mask was least robust to shifts in LBIDD.

## 8 DISCUSSION

We showed that models leveraging informative missingness can achieve equally robust prediction under ignorable and even some non-ignorable shifts, often outperforming unbiased methods. This challenges the common belief that only unbiased estimators, like multiple imputation, are reliable (Steyerberg & Vergouwe, 2014; Wynants et al., 2020; Ipsen et al., 2022). Our theoretical findings are quite general: if missingness is ignorable in both environments, any method that recovers the optimal predictor for the source environment will also fare well in the target environment.

Rather than a clear preference for unbiased methods, this suggests that different estimators might be robust to different types of shift. Observing that some estimators may have high error rates when missingness decreases in the target environment, we proposed NeuMISE, a novel method that showed robust performance under those conditions. The use of unbiased methods like MICE, on the other hand, was most beneficial when missingness was extremely informative. In those cases, MICE prevented the model from learning a strong signal that was no longer present in the target environment. As a result, MICE appeared worse in the source environment but resulted in considerably more robust models. More generally, determining the — potentially method-specific — factors that lead to a good estimation appears promising; we leave this to future work.

We tested our results on simulated and real-world covariates across a range of missingness mechanisms. For computational reasons, we limited ourselves to a single, non-linear outcome function. However, while additional experiments including further datasets, models, missingness mechanisms, shifts, and outcomes may lead to different conclusions about the relative performance of models, we believe that the main conclusion will remain unchanged: whether or not a prediction model is robust to changes in missingness is not necessarily linked to its ability to recover the complete data predictor.

## ETHICS STATEMENT

This study was conducted in accordance with the ethical guidelines and best practices laid out by the International Conference on Learning Representations (ICLR). We reviewed the ethical implications of our study and determined that there are no foreseeable risks or harms to individuals or communities. Some of the data used in this study describes individual patients from the 1995 Linked Birth and Infant Death Data (LBIDD). LBIDD is published by the US National Center for Health Statistics and contains yearly birth certificate data for each infant born in the United States, Puerto Rico, The Virgin Islands, and Guam. For each infant under 1 year of age who dies, the data was linked to information from death certificate by the data provider. To ensure privacy, LBIDD only contains fully anonymized information. No personally identifiable information were available to the authors or used in the study. We did not attempt to link this dataset with data from other sources. All data was accessed through the US National Vital Statistics System using an openly available link: https://www.cdc.gov/nchs/nvss/linked-birth.htm. Usage in this study complied with the terms and conditions set forth by the data provider. Given the nature of the data and the research, no formal ethical approval was sought.

## REPRODUCIBILITY STATEMENT

In line with the commitment to transparent science, we have taken the following steps to ensure that the experiments and results presented in this paper can be reproduced. The main text and appendix aim to give a comprehensive description of the experimental setup, including data generation steps, model architectures, hyperparameter ranges and final hyperparameters, and training and evaluation procedures. For full reproducibility, the code for all experiments is provided as an anonymised ZIP archive and will be made available as a GitHub repository following the review period. The code includes documentation and comments to facilitate understanding and replication of the experiments. All experiments were performed on CPUs using Python 3.10 for Linux. We additionally tested our code on an Apple M1 Max with Ventura 13.2.1. Details on packages and versions can be found in a conda environment file in the ZIP archive. While every effort has been made to ensure the reproducibility of the results, some variability due to hardware or software differences may be expected.

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

# A  EXTENDED THEORETICAL RESULTS

## A.1  PROOF OF THEOREM 1

**Theorem 1 (Equivalence of Bayes predictors under missingness shift)** *Assume that the data is generated according to Equation (2). Then, if there is a missingness shift, $\forall m \in \{0,1\}^d$ : $\tilde{f}^{\star}{}_m = \tilde{g}^{\star}{}_m$ holds in general only if the missingness shift is ignorable.*

From Equation (4) we know that the Bayes predictor depends on the complete data model $f^*$ and the conditional probability distribution $p(X_{mis} \mid X_{obs}, M)$ of the unobserved data. By assumption, $f^*$ is not affected by a missingness shift and remains fixed across environments. Any changes in missingness may therefore effect the Bayes predictor only through $p(X_{mis} \mid X_{obs}, M)$, which may be refactored as

$$
\begin{aligned}
p(X_{mis} \mid X_{obs}, M) &= \frac{p(X_{mis}, X_{obs}, M)}{p(X_{obs}, M)} \\
&= \frac{p(X_{mis}, X_{obs})p(M \mid X_{mis}, X_{obs})}{p(X_{obs})p(M \mid X_{obs})}.
\end{aligned}
\tag{5}
$$

We can see that this probability depends on $M$ unless the missingness mechanism is ignorable, i.e., $p(M \mid X) = p(M \mid X_{obs})$ (Rubin, 1976). Therefore, if the missingness shift is ignorable we have:

$$
p(X_{mis} \mid X_{obs}, M) = \frac{p(X_{mis}, X_{obs})}{p(X_{obs})} = \frac{q(X_{mis}, X_{obs})}{q(X_{obs})} = q(X_{mis} \mid X_{obs}, M),
\tag{6}
$$

where the second equality follows from the assumption that $p(\cdot) = q(\cdot)$ in the absence of missingness.

However, if there is shift in missingness between train and test time, according to Definition 1, we have $p(M \mid X_{obs}, X_{mis}) \neq q(M \mid X_{obs}, X_{mis})$ and therefore — in the absence of ignorability in the missingness shift — in general also $p(X_{mis} \mid X_{obs}, M) \neq q(X_{mis} \mid X_{obs}, M)$. For a non-constant outcome function $f^*(X) \neq c$ and a randomly chosen shift in missingness, we get $\tilde{f}^{\star}{}_m \neq \tilde{g}^{\star}{}_m$ unless both the missingness in the source and target environment are ignorable.

## A.2  GENERALITY OF RESULTS

In the main body of the paper as well as the proof above, we make the assumption that the outcome separates into a deterministic function $f^{\star}(X)$ of the full covariates and an additive noise $\epsilon$ with $\mathbb{E}[\epsilon \mid X_{obs}, M] = 0$. This assumption simplifies Equation (4), leaving only the complete data function $f^{\star}(X)$ and the imputation model $p(X_{mis} \mid X_{obs}, M)$. While we believe that this simplification helps in understanding the main arguments of the paper, it is not strictly needed. To see why, we can formulate the problem in terms of probabilities rather than expected values:

$$
p(Y|X_{obs}, M) = \int_{X_{mis}} p(Y|X_{mis}, X_{obs}, M)p(X_{mis} \mid X_{obs}, M)dX_{mis}.
\tag{7}
$$

Given $M \perp\!\!\!\perp Y, X_{mis} \mid X_{obs}$, Equation (7) again simplifies as

$$
p(Y|X_{obs}, M) = \int_{X_{mis}} p(Y|X_{mis}, X_{obs})p(X_{mis} \mid X_{obs})dX_{mis},
\tag{8}
$$

and the proof can proceed as above while leaving both the outcome distribution (e.g., Bernoulli) and the noise unspecified. Note that the conditional independence $M \perp\!\!\!\perp Y \mid X_{obs}$ is introduced by our assumption that $p(M \mid X, Y) = p(M \mid X_{obs})$.

### A.3 NOTE ON CHANGES IN BAYES RISK UNDER SHIFTS

The equivalence of the Bayes predictor introduced in Theorem 1 does not imply equal model performance in the source and target environment. To see why, we consider the Bayes risk $\mathcal{R}^\star$ — i.e., the average risk of the Bayes predictor (Le Morvan et al., 2021). Assuming all Bayes predictors $\mathcal{F}$ are known, we calculate the Bayes risk for a dataset $\mathcal{D}$ of size $N$ as

$$\hat{\mathcal{R}}^\star_\mathcal{D} = \frac{1}{N} \sum_{i=1}^N \left( y_i - \tilde{f}^\star_{m_i}(x_{obs,i}) \right)^2. \tag{9}$$

Unless the risks for all $f_m \in \mathcal{F}$ are equal, i.e., $\forall m \in \{0,1\}^d : \mathcal{R}(f_m) = c$ with $c \in \mathbb{R}$, the risk $\hat{\mathcal{R}}^\star_\mathcal{D}$ will depend on the relative frequency of each missingness pattern $m$ in $\mathcal{D}$ and — since a missingness shift may affect this relative frequency — may change even when $\mathcal{F}$ remains unchanged. Depending on the shift in missingness, a model's performance may therefore degrade and no longer be fit for purpose even if it correctly estimated all Bayes predictors in the target environment.

### A.4 ROBUST ESTIMATION UNDER Y-DEPENDENT MISSINGNESS

Section 6.2 discusses how $Y$-dependence in the source environment may still admit the estimation of a robust estimator if missingness in the target environment is ignorable and no longer depends on $Y$. This can be easily seen for cases where data is always fully observed in the target environment. In fact, conditional imputation on $Y$ has been used for a long time to derive $\tilde{g}^\star_{[0,...,0]}$, the Bayes predictor of the complete data.

Our results suggest that identifiability should not be restricted to $\tilde{g}^\star_{[0,...,0]}$ but should apply to all Bayes predictors $\tilde{g}^\star_m \in \mathcal{G}$ of the target environment. This conclusion is based on the following observations:

1. Even under $Y$-dependent missingness in the source environment, the underlying conditional distribution of $X_{mis}$ given $X_{obs}$ does not change between environments, i.e., $p(X_{mis} \mid X_{obs}) = q(X_{mis} \mid X_{obs})$. This is true by assumption.

2. The joint distribution $p(Y, X_{mis}, X_{obs})$ is identifiable in the source environment, since missingness is ignorable conditional on $Y$.

3. The conditional distribution $p(X_{mis} \mid X_{obs})$ may be derived from the joint distribution as $\int_Y p(Y, X_{mis}, X_{obs}) dY$.

Based on the above, we could imagine the following imputation-based strategy to estimate a Bayes predictor for the target environment:

1. Estimate an imputation model $\hat{p}(X_{mis} \mid X_{obs}, Y)$ from the source data.

2. Use $\hat{p}(X_{mis} \mid X_{obs}, Y)$ to generate a fully observed dataset $\hat{\mathcal{D}}^\star$ by filling in the missing values with one or more random draws from the imputation model.

3. Use $\hat{\mathcal{D}}^\star$ to estimate $\hat{f}^\star(X)$ as well as a second imputation model $\hat{q}(X_{mis} \mid X_{obs})$ for the target environment. Due to $\hat{\mathcal{D}}^\star$ being fully observed, $\hat{q}(X_{mis} \mid X_{obs})$ will be an unbiased estimate of $q(X_{mis} \mid X_{obs})$.

Through the use of established imputation methods like MICE, we believe that the strategy outlined above results in a consistent estimator of the target environment's Bayes predictors $\tilde{g}^\star_m \in \mathcal{G}$. However, it is unclear how uncertainty in $\hat{p}(X_{mis} \mid X_{obs}, Y)$ may affect fidelity of $\hat{q}(X_{mis} \mid X_{obs})$. We leave this to future work.

### A.5 IPW FOR DOMAIN ADAPTATION UNDER MISSINGNESS SHIFT

Zhou et al. (2023) proposed to re-weight samples with $q(x)/p(x)$ in order to account for the covariate shift induced by an ignorable missingness shift with missing indicators. Re-weighting of

this form is a common approach in domain adaptation with shared support. Written in terms of the observed data vector $\tilde{X}$, we have:

$$\frac{q(\tilde{X})}{p(\tilde{X})} = \frac{q(X_{obs}, M)}{p(X_{obs}, M)} = \frac{q(X_{obs})q(M|X_{obs})}{p(X_{obs})p(M|X_{obs})} = . \tag{10}$$

Since by assumption $p(X_{obs}) = q(X_{obs})$, we get

$$\frac{q(M|X_{obs})}{p(M|X_{obs})}. \tag{11}$$

Interestingly, if we do not have information on missingness in the target environment $q(M|X_{obs})$, this approach reduces to the well-known Inverse Probability Weighting (IPW) for missing data Seaman & White (2013), although probability weights are calculated for all samples and not just those with completely observed covariates.

## B  EXTENDED EMPIRICAL RESULTS

This section contains additional empirical results that were cut from the main body due to space limitations. They include main experiment results for MCAR missingness (Figure 5), complete data results for non-monotone MAR and MNAR (Gaussian self-masking) missingness (Figure 6), and results for shifts from lower levels of missingness (25%) in the source environment to higher levels of missingness (50%) in the target environment (Figure 7), essentially reversing the experiments presented in the main body.

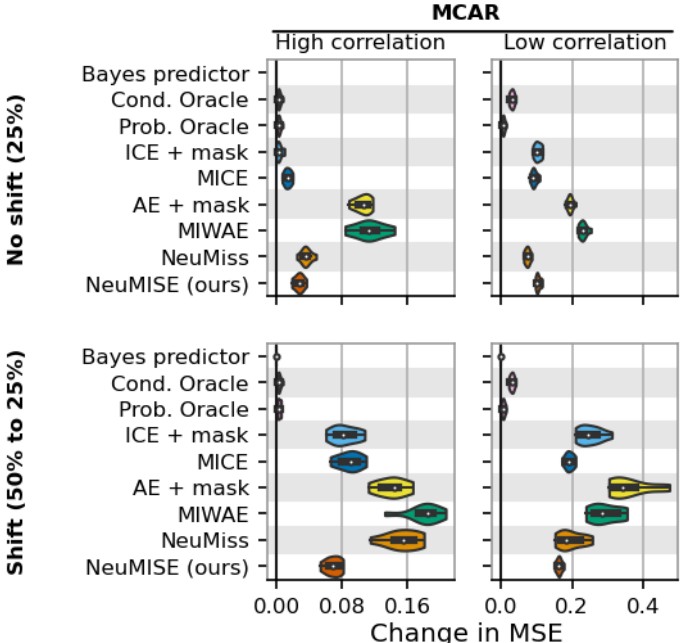

Figure 5: MSE in the target environment (25% missingness) compared to the performance of the Bayes predictor. We evaluated estimators trained within the same environment (no shift) and estimators trained in another source environment (shift from 50% missingness).

**Results for MCAR missingness**   were very similar to the results obtained for non-monotone MAR 5). This isn't particularly surprising, given that both mechanisms allow for unbiased estimation of the conditional probability of the missing data (see Appendix C for a detailed description)

**Results for complete data** are described in the main body (Figure 6).

**Results for increasing missingness in the target environment** show a higher variability — and hence potential instability — of NeuMISE in this setting (Figure 7). ICE+mask and MICE, on the other hand, perform particularly well in this setup. Due to a lower variability in the iterative procedure, they may be able to arrive at a better imputation model in data with lower levels of missingness.

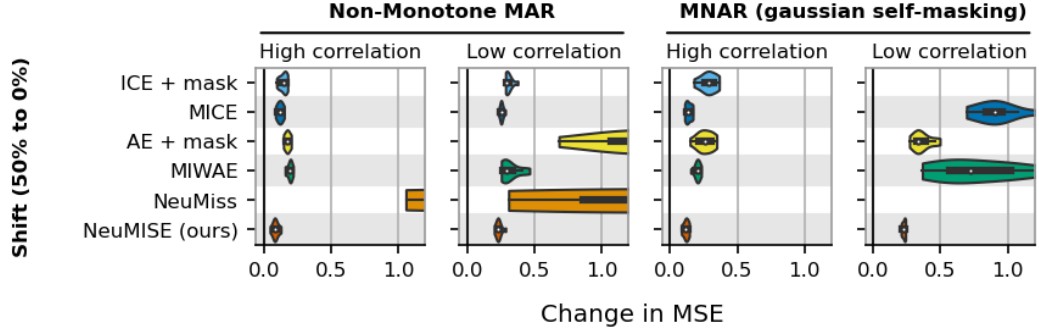

Figure 6: MSE in complete data compared to the performance of the complete data predictor. Estimators were trained with 50% missingness. Oracles perform ideal by definition and were omitted. Where results are not visible, estimators performed worse and the plot was clipped for readability.

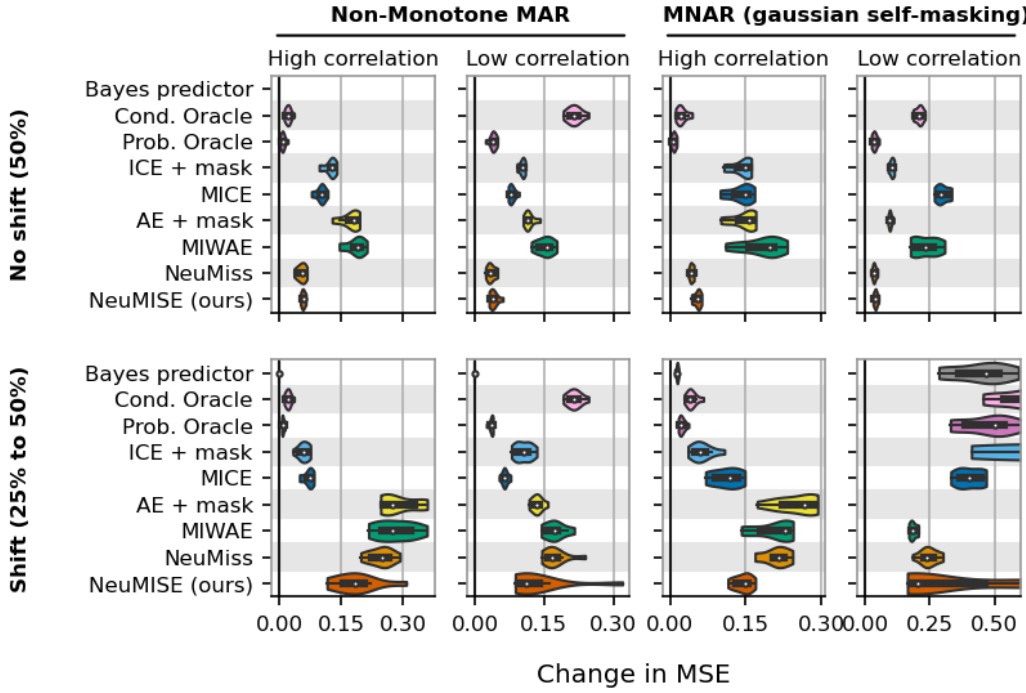

Figure 7: MSE in a target environment with *more* missingness (50% missingness), compared to the performance of the Bayes predictor. We evaluated estimators trained within the same environment (no shift) and estimators trained in another source environment (shift from 25% missingness). Where results are not visible, estimators performed worse and the plot was clipped for readability.

## C  MISSINGNESS MECHANISMS

In general, the missingness indicator $M$ may depend on the entire feature set $(X_{obs}, X_{mis})$. That is, the probability of observing/missing a data point may depend on all observed and missing features, including itself. In this general form, the missingness mechanism is denoted by the irreducible conditional density $p_\phi(M \mid X_{obs}, X_{mis})$ where $\phi$ parameterizes $p$. Without further assumptions, this density is not identifiable from available data due to the dependence on unobserved $X_{mis}$. To simplify the density into an identifiable expression, Rubin (1976) categorizes possible assumptions on the missing data model in the form of independence relations between $M$ and $X$:

**Missing Completely At Random (MCAR)**  In MCAR, $M$ is independent of both observed and missing components. Consequently, for all $m \in \{0, 1\}^d$ we have $p(M = m \mid X_{obs}, X_{mis}) = p(M = m)$. Although MCAR conveniently allows for an unbiased estimation via straightforward solutions (e.g. complete-case analysis), it makes the strong assumption that missingness is a purely random event. In many real-world situations, this is rather unrealistic and might oversimplify the underlying causes of missingness.

**Missing At Random (MAR)**  In MAR, $M$ is independent of the unobserved data only given the observed components. This simplifies the probability of missingness to $p(M = m \mid X_{obs}, X_{mis}) = p(M = m \mid X_{obs})$ for all $m \in \{0, 1\}^d$. Even though the assumptions underlying MAR are still strong, they are less stringent compared to MCAR and much more plausible in practice. As a result, MAR is the default assumption for many existing methods (Le Morvan et al., 2021).

**Missing Not At Random (MNAR)**  Data is said to be MNAR, if neither MCAR nor MAR holds, that is, if missingness may depend on both observed and unobserved data. Hence, the mechanism is given by the irreducible form $p(M = m \mid X_{obs}, X_{mis})$. In general, MNAR mechanisms are nonignorable and unidentifiable. However, many resolving assumptions have been introduced in the literature which may prove efficient in certain circumstances (see for example Malinsky et al. (2022); Chen (2022); Li et al. (2022)).

### C.1  NOTE ON COMPLETE-CASE ANALYSIS IN X-DEPENDENT MISSINGNESS

Complete-case analysis results in unbiased estimation of the complete data predictor $f^\star$ as long as missingness solely relies on covariates $X$, i.e., the missingness indicator $M$ is conditionally independent of $Y$ given $X$ (White & Carlin, 2010). This is not only true for MCAR missingness — for which complete-case analysis is always unbiased (Rubin, 1976) — but also for MAR and even MNAR missingness. Although this fact was established early on in the missing data literature, it may come as surprise to many, not the least because it does not fit neatly into the MCAR/MAR/MNAR categorisation (White & Carlin, 2010). The implication for prediction modelling is that even otherwise biased estimators may be able to estimate complete data predictor $f^\star$ if missingness does not depend on $Y$ or unobserved factors related to $Y$, provided sufficient complete (or almost complete) samples are available in the training dataset.

## D  DETAILED EXPERIMENTAL SETUP

### D.1  SIMULATED COVARIATES

In all experiments that used fully simulated data, we simulated $N = 100,000$ observations of $d = 50$ covariates from a multivariate normal distribution as $X \sim N(\mu, \Sigma)$, with $\mu \in \mathbb{R}^d$ and $\Sigma \in \mathbb{R}^{d \times d}$. The means $\mu \sim N(0, 1)$ were drawn from a standard normal distribution. The covariance $\Sigma = BB^T$ was defined as the square of a matrix $B \in \mathbb{R}^{d \times \lceil \lambda d \rceil}$, where $\lambda$ governed the degree of correlation between covariates. Every entry of $B$ independently followed a standard normal distribution. In this setup, lower $\lambda$ implies higher correlation. We used $\lambda = 0.7$ and $\lambda = 0.3$ to simulate high and low correlation settings, respectively. For every one of the ten repetitions, we simulated a separate, independent dataset.

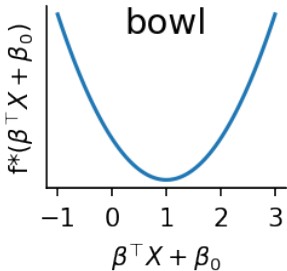 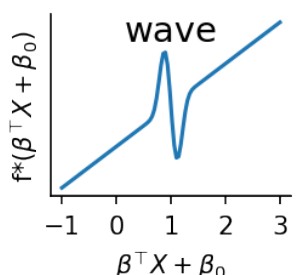 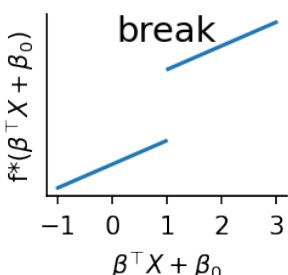

Figure 8: Candidate outcome functions (reproduced from Le Morvan et al. (2021)). We chose the wave function (middle figure) for all of our experiments.

## D.2 LINKED BIRTH AND INFANT DEATH DATA (LBIDD)

The LBIDD (MacDorman & Atkinson, 1998) contains the yearly natality and mortality data for infants born and deceased in U.S. territories since 1995. In our experiments on real-world data, we used a processed subset of the covariate data from the LBIDD of 1995. Specifically, we took data from the denominator file, which contains information of approximately 3,900,000 infants.

Out of the total of ∼170 variables available in LBIDD, we selected 50 covariates based on their significance in predicting mortality (MacDorman & Atkinson, 1998) and the percentage of missing observations (ranging from 0% to 1.25% in the chosen variables). To obtain a complete dataset, we removed the observations with any missing values, accounting for the 4.34% of the total data. Although this procedure may have introduced a slight bias in the covariate distribution, we can disregard it as we simulated the outcome and were not concerned with the actual relationships. Nevertheless, we noted that for continuous variables there was a noticeable reduction in variance among the complete cases compared to the raw data (Table 1). This can be attributed to the co-occurrence of missing values in one covariate and out-of-bound values in another covariate within the same observation. For the same reason, and due to the mean's sensitivity to outliers, there was a significant shift in the mean of 'Previous births and interruptions'. In contrast, the median, which is robust to outliers, shows minimal change (for 'Weight at birth in grams' it shifts from 3373 to 3374 while for 'Mother's age','Month parental care began','Weeks of gestation' and 'Prior births and interruptions' it doesn't change, taking values 27, 1, 39 and 2 respectively).

We also recoded the variables to ensure a consistent representation and a format more suited for the experiments. The resulting dataset comprises both continuous and binary variables. In our experiments, we worked with a randomized subsets of this dataset.

## D.3 OUTCOME

For both the fully simulated covariates and the LBIDD covariates, we generated a continuous outcome $Y = h(X) + \epsilon$, where $h$ is the outcome function and $\epsilon \sim N(0, \sigma_\epsilon)$ represents random Gaussian noise. We chose $\sigma_\epsilon$ to ensure a signal-to-noise ratio of 10. In previous work, Le Morvan et al. (2021) considered three nonlinear forms for $h$ (Figure 8). Due to computational constraints, we limited our analysis to a single one of those functional forms, the wave function, which we believe provides reasonable complexity and variability:

$$h(X) = (X\beta + \beta_0 - 1) + \sum_{(a_i, b_i) \in S} a_i \Phi(\gamma(X\beta + \beta_0 + b_i)),$$

where $\Phi$ is the standard Gaussian cdf, $\beta$ is a vector of linear coefficients, $\gamma$ is a scalar governing the curvature, and $S = \{(a_i, b_i)\}$ are parameters determining the amplitude and location of the waves. To keep our experiments in line with Le Morvan et al. (2021), $\beta$ was chosen as a vector of ones rescaled so that $var(X\beta) = 1$, $\gamma$ was set to $\gamma = 20\sqrt{\pi/8}$, and $S$ was defined as $S = \{(2, -0.8), (-4, -1), (2, -1.2)\}$.

Table 1: Description of the 50 variables extracted from the Linked Birth and Infant Death Data (LBIDD).

| Variable | Range | Percentage of missing values | Original distribution | Final distribution |
|---|---|---|---|---|
| Infant is male | 0/1 | 0.00% | 51.19% | 51.18% |
| Mother's age | 10 - 49 | 0.00% | 26.85±6.11 | 26.84±6.10 |
| Mother is married | 0/1 | 0.00% | 67.86% | 68.02% |
| Month prenatal care began | 01 - 09 | 0.00% | 1.33±0.80 | 1.31±0.75 |
| Weight at birth in grams | 0227 - 8165 | 0.00% | 3330±621 | 3329±602 |
| Plurality of births | 0/1 | 0.00% | 2.61% | 2.58% |
| Birth in the flue season | 0/1 | 0.00% | 65.11% | 65.14% |
| Birth on the weekend | 0/1 | 0.00% | 22.71% | 22.67% |
| Deceased in 1995 | 0/1 | 0.00% | 0.64% | 0.60% |
| Mother is black | 0/1 | 0.00% | 15.46% | 15.43% |
| Delivered not at the hospital | 0/1 | 0.02% | 1.02% | 0.96% |
| Attendant at delivery not M.D. | 0/1 | 0.18% | 10.20% | 10.21% |
| Mother from the US | 0/1 | 0.25% | 81.43% | 81.62% |
| Delivered by c-section | 0/1 | 0.75% | 20.84% | 20.82% |
| Amniocentesis | 0/1 | 0.83% | 3.2% | 3.16% |
| Electronic fetal motoring | 0/1 | 0.83% | 81.24% | 81.53% |
| Induction of labor | 0/1 | 0.83% | 15.99% | 16.09% |
| Stimulation of labor | 0/1 | 0.83% | 16.09% | 16.14% |
| Tocolysis | 0/1 | 0.83% | 1.89% | 1.88% |
| Ultrasound | 0/1 | 0.83% | 61.14% | 61.54% |
| Other obstetric procedures | 0/1 | 0.83% | 5.05% | 5.51% |
| Weeks of gestation | 17 - 47 | 0.94% | 39.51±6.35 | 38.96±2.61 |
| Prior births and interruptions | 01 - 40 | 0.94% | 3.32±9.46 | 2.40±1.56 |
| Fever at labor | 0/1 | 0.98% | 1.60% | 1.58% |
| Moderate & high meconium | 0/1 | 0.98% | 5.71% | 5.68% |
| Premature rupture of membrane | 0/1 | 0.98% | 3.06% | 3.04% |
| Abruptio placenta | 0/1 | 0.98% | 0.57% | 0.56% |
| Placenta previa at labor | 0/1 | 0.98% | 0.34% | 0.33% |
| Excessive bleeding in labor | 0/1 | 0.98% | 0.58% | 0.57% |
| Seizure during labor | 0/1 | 0.98% | 0.04% | 0.04% |
| Precipitous labor | 0/1 | 0.98% | 1.91% | 1.91% |
| Prolonged labor | 0/1 | 0.98% | 0.88% | 0.87% |
| Cephalopelvic disproportion | 0/1 | 0.98% | 2.54% | 2.51% |
| Cord prolapse | 0/1 | 0.98% | 0.23% | 0.23% |
| Other complication | 0/1 | 0.98% | 13.60% | 13.60% |
| Cardiac disease in mother | 0/1 | 1.17% | 0.48% | 0.48% |
| Lung disease in mother | 0/1 | 1.17% | 0.69% | 0.69% |
| Diabetes in mother | 0/1 | 1.17% | 2.52% | 2.51% |
| Pregnancy-related hypertension | 0/1 | 1.17% | 3.41% | 3.41% |
| Medical risk factor, other | 0/1 | 1.17% | 13.48% | 13.49% |
| Hydramnios in mother | 0/1 | 1.17% | 1.14% | 1.13% |
| Eclampsia in mother | 0/1 | 1.17% | 0.37% | 0.37% |
| Hemoglobinopathy in mother | 0/1 | 1.17% | 0.07% | 0.07% |
| Incomplete cervix in mother | 0/1 | 1.17% | 0.24% | 0.23% |
| Chronic hypertension in mother | 0/1 | 1.17% | 0.67% | 0.67% |
| Previous preterm birth | 0/1 | 1.17% | 1.14% | 1.13% |
| Anemia | 0/1 | 1.25% | 0.11% | 0.11% |
| Hyaline membrane disease | 0/1 | 1.25% | 0.67% | 0.67% |
| Meconium aspiration syndrome | 0/1 | 1.25% | 0.24% | 0.24% |
| Seizures | 0/1 | 1.25% | 0.09% | 0.09% |

*Range*: For binary variables, it is represented as '0/1', where an 1 signifies presence. For numerical variables, it indicates the range of possible values.

*Original and Final Distribution*: Summary of values before (original) and after (final) excluding rows with missing values in the raw data. For binary variables, this represents the percentage of observations with the value 1. For numerical variables, it shows the 'mean ± standard deviation'.

### D.4 MISSINGNESS

**MCAR** To simulate experiments under the MCAR assumption, we masked each covariate randomly and independently with a constant probability. We defined the probability of missingness for a covariate $j \in \{1, \ldots, d\}$ as

$$P(M^{(j)} = 1) = P(U^{(j)} < p),$$

where $U^{(j)} \sim Uniform(0, 1)$ and $p \in [0, 1]$ is the desired proportion of missing values.

**Non-monotone MAR** We simulated experiments under the non-monotone MAR assumption as follows: First, we randomly selected a subset of 30% of the covariates which were masked according to MCAR. Then, based on the observed values per sample, the remaining covariates were masked using a logistic function.

More precisely, let $J$ be the set of variables masked with MCAR and $L$ the set of variables masked using a logistic function with $|J| = \lfloor d * 0.3 \rfloor$ and $|L| = d - |J|$. The probability of a covariate $j \in J$ being missing was defined as

$$P(M^{(j)} = 1) = P(U^{(j)} < p),$$

where $U^{(j)} \sim Uniform(0, 1)$ and $p \in [0, 1]$ is the desired proportion of missing values. After all variables in $J$ were masked, the probability of a covariate $l \in L$ being missing was conditionally defined as

$$P(M^{(l)} = 1 \mid X_{obs}^{(J)}) = P\left(U^{(l)} < ps^{(l)}(X_{obs}^{(J)})\right)$$

where $X_{obs}^{(J)}$ are all the covariates in $J$ that were left unmasked after applying MCAR missingness. The logistic function for covariate $l$ was defined as

$$ps^{(l)}(X_{obs}^{(J)}) = sigmoid\left(X_{obs}^{(J)}\gamma^{(l)} + \gamma_0^{(l)}\right),$$

with coefficients $\gamma^{(l)} \in \mathbb{R}^{|J|}$ obtained as

$$\gamma^{(l)} = \frac{\delta^{(l)}}{s^{(l)}v^{(l)}}$$

where $\delta^{(l)} \in \mathbb{R}^{|J|}$ and $s^{(l)}, v^{(l)} \in \mathbb{R}$ with

$$\delta^{(l)} \sim \mathcal{N}(0, 1),$$
$$s^{(l)} \sim Uniform(0.1, 0.5),$$
$$v^{(l)} = \sqrt{(\delta^{(l)})^T \Sigma_{obs} \delta^{(l)}}.$$

$v^{(l)}$ scales the coefficients on the logit scale proportionally to the covariance $\Sigma_{obs}$ of the observed variables. Finally, the intercept $\gamma_0^{(j)}$ was determined in a manner that guarantees the intended proportion of missingness $p$. Therefore, the missingness distribution was parameterized by the proportion of missingness $p$ and the intercept $\gamma_0$ and slopes $\gamma$ of the logistic function.

**MNAR (gaussian self-masking)** We simulated experiments under MNAR mechanism using Gaussian self-masking, which was also used by Le Morvan et al. (2020). Let $\tilde{\mu}$ and $\tilde{\sigma}$ denote the means and standard deviations of a Gaussian density function. The probability of a covariate $j \in \{1, \ldots, d\}$ being missing then depends on this function and its underlying value as:

$$p(M^{(j)} = 1 \mid X^{(j)}) = K^{(j)}exp\left(-\frac{1}{2}\frac{(X^{(j)} - \tilde{\mu}^{(j)})^2}{(\tilde{\sigma}^{(j)})^2}\right).$$

Table 2: Parameter choices governing missingness in source and target environments. Parameters are denoted $a$ in the source environment and $a'$ in the target environment. $p'(\cdot)$ conforms to $q(\cdot)$ in the main text.

| Mechanism | Source | Target |
|---|---|---|
| MCAR | $p(M^{(j)} = 1) = 0.5$ | $p'(M^{(j)} = 1) = 0.25$ |
| Non-monotone MAR | $p(M^{(j)} = 1) = 0.5$ $\gamma^{(l)} = \delta^{(l)}/(s^{(l)}v^{(l)})$ | $p'(M^{(j)} = 1) = 0.25$ $\gamma'^{(l)} = \delta'^{(l)}/(s'^{(l)}v'^{(l)})$ |
| Gaussian self-masking | $p(M^{(j)} = 1) = 0.5$ $k = 2$ | $p'(M^{(j)} = 1) = 0.25$ $k' = 2$ |
| MAR-Y | $p(M^{(j)} = 1) = 0.5$ $\gamma^{(l)} = \delta^{(l)}/v^{(l)}$ | $p'(M^{(j)} = 1) = 0.25$ $\gamma'^{(l)} = \delta'^{(l)}/v'^{(l)}$ |

The values of $\tilde{\mu}^{(j)}$, $\tilde{\sigma}^{(j)}$, and $K^{(j)}$ are governed by a hyperparameter $k$ and the desired marginal rate of missingness. For $k > 0$, higher values are more likely to go missing whereas for $k < 0$, lower values are more likely to go missing. In our experiments, we chose a value of $k = 2$ in both environments. The multivariate missingness is given by $p(M \mid X) = \prod_{j=1}^{d} p(M^{(j)} \mid X^{(j)})$. See Le Morvan et al. (2020) and Le Morvan et al. (2021) for a detailed discussion of Gaussian self-masking.

**MAR-Y** We simulated experiments that dependet on the outcome $Y$ by masking covariates based on a logistic function of $Y$:

$$P(M^{(j)} = 1 \mid Y) = P\left(U^{(j)} < ps^{(j)}(Y)\right),$$

where $U^{(j)} \sim Uniform(0,1)$ and $p \in [0,1]$ is the desired proportion of missing values. The logistic function for covariate $j \in \{1, \ldots, d\}$ was defined as

$$ps^{(j)}(Y) = sigmoid\left(Y\gamma^{(j)} + \gamma_0^{(j)}\right)$$

with coefficient $\gamma^{(j)} \in \mathbb{R}$ obtained as

$$\gamma^{(j)} = \frac{\delta^{(j)}}{v^{(j)}}$$

where $\delta^{(j)}, v^{(j)} \in \mathbb{R}$ with

$$\delta^{(j)} \sim \mathcal{N}(0,1),$$
$$v^{(j)} = \delta^{(j)}\sigma_Y.$$

$v^{(j)}$ again scales the coefficients on the logit scale proportionally to the standard deviation $\sigma_Y$ of the outcome while $\gamma_0^{(j)}$ still guarantees the intended proportion of missingness $p$.

D.5  SHIFTS

Our main experiment investigated the effects of a shift from a higher level of missingness to moderate missingness, which may for example result from more diligent recording of information after model deployment. **No shift:** To evaluate prediction performance in the absence of a missingness shift, both the source and target environments were generated using the same missingness mechanism and a missingness probability of 25%. **Missingness shift:** To simulate missingness shift, the probability of missingness was changed from 50% in the source environment to 25% or 0% in the target environment (Table 2). For the monotonous MAR and MAR+Y, the parameters of the logistic function (i.e., intercept $\gamma_0^{(j)}$ and slopes $\gamma^{(j)}$) were newly drawn and could vary between source and

target environment. We repeated our analysis for a setup where missingness increased from 25% to 50%.

# E  ESTIMATORS

## E.1  BAYES PREDICTOR

The Bayes predictor can be derived analytically for special cases. For example, the Bayes predictor for linear prediction of multivariate normal covariates with *M(C)AR* missingness is given as

$$\tilde{f}^{\star}(X_{obs}, M) = \beta_0 + X_{obs}\beta_{obs} + (\mu_{mis} + \Sigma_{mis,obs}(\Sigma_{obs})^{-1}(X_{obs} - \mu_{obs}))\beta_{mis}, \quad (12)$$

where $\beta$ are the coefficients of the linear model, and $\mu$ and $\Sigma$ are the mean and covariance of the data distribution (Le Morvan et al., 2020). Notably, Equation (12) does not depend on any parameters of the missingness mechanism and is left unchanged by a shift to another M(C)AR mechanism. Although MNAR, the Bayes predictor of multivariate normal covariates with Guassian self-masking can also be analytically derived and is given as

$$\tilde{f}^{\star}(X_{obs}, M) = \beta_0 + X_{obs}\beta_{obs} + (Id + D_{mis}\Sigma_{mis|obs}^{-1})^{-1} \times (\tilde{\mu}_{mis} + D_{mis}\Sigma_{mis|obs}^{-1}A)\beta_{mis}, \quad (13)$$

where $D$ is a diagonal matrix such that $diag(D) = (\tilde{\sigma}_1^2, ..., \tilde{\sigma}_s^2)$ and $A = \mu_{mis} + \Sigma_{mis,obs}(\Sigma_{obs})^{-1}(X_{obs} - \mu_{obs})$ is the conditional expectation from Equation (12) (Le Morvan et al., 2020). Unlike the M(C)AR predictor, Equation (13) does depend on the parameterisation of the missingness mechanism through $D$ and thus changes under shifts. Le Morvan et al. (2021) extended Equations (12) and (13) to the non-linear outcome function considered in our experiments.

## E.2  ORACLES

We considered two distinct oracles that have access to the true imputation function as well as the true complete data predictor $f^{\star}$, but differ in the way they perform imputation.

**The Conditional Oracle**   uses conditional expectation to impute the data, which is then used by the true outcome function $f^{\star}$. The conditional imputation function $\Phi^{CI} \in (\mathbb{R} \cup \{\texttt{NA}\})^d \to \mathbb{R}$ for any covariate $j \in \{1, ..., d\}$ is defined as

$$\Phi_j^{CI}(\tilde{X}) = \begin{cases} X^{(j)} & \text{if } M^{(j)} = 0, \\ \mathbb{E}[X^{(j)} \mid X_{obs}, M] & \text{if } M^{(j)} = 1. \end{cases}$$

The Conditional Oracle performs well if there is only little variation in the data or if the covariates are highly correlation, in which cases the expected value is close to the true value (Le Morvan et al., 2021). On the other hand, the performance of the Conditional Oracle may be limited if there is a low correlation between covariates, since the expected value ignores the variation in the data.

**The Probabilistic Oracle**   uses the full conditional probability distribution to impute the missing data, which is then used by $f^{\star}$. The following equation is employed to perform a prediction:

$$\hat{f}(Y|X_{obs}) = \int f^{\star}(X_{obs}, X_{miss})p(X_{miss}|X_{obs}, M = m)dX_{miss}. \quad (14)$$

The Probabilistic Oracle overcomes the limitation of the Conditional Oracle, by drawing samples from the conditional probability distribution of the missing data given the observed data and the missingness pattern $p(X_{miss}|X_{obs}, M = m)$, reflecting the variation in the data. In the limit of infinite samples from the conditional probability, the Probabilistic Oracle equals the Bayes predictor. To allow for a fair comparison with learned estimators that were only computationally feasible for a low number of draws, we approximate this with $n_{draws} = 5$ in our experiments.

### E.3 ITERATIVE ESTIMATORS

**Imputation by Chained Equations (ICE) + mask** consists of a concatenation of imputations from the ICE algorithm (Van Buuren, 2018) with the missingness indicator $m$ (the "mask"), followed by a regression via multilayer perceptron (MLP). We used the Scikit-learn's conditional imputer `IterativeImputer` with `BayesianRidge` regression for imputation. This imputer first initializes each missing covariate with the mean value of its observed entries. Each missing covariate is then iteratively imputed by regressing on all other (preliminary filled) covariates. This process is repeated for all missing covariates over a number of iterations, which can be specified as a parameter. In our experiments, we used the default value of $n_{iter} = 10$. By not sampling from the posterior distribution, imputation using ICE is limited to the conditional expectation, without modelling the uncertainty associated with the imputed values. Concatenating the mask, and thus using the missingness information, allows ICE to more easily account for MNAR settings.

**Multiple Imputation by Chained Equations (MICE)** Similar to ICE, MICE iteratively imputes the missing covariates via `BayesianRidge` regression, but instead of imputing with the conditional expectation, it samples from the conditional distribution. By drawing $n_{draws} = 5$ values from the conditional distribution, it introduces variability into the imputation process and generates multiple imputed datasets. This allows it to approximate the integral in Equation (4) and account for the uncertainty introduced by missing values. For MICE, we did not concatenate the mask, but use the complete data only as an input for the MLP, aiming to obtain an unbiased estimate of the complete data function $f^\star$ from the partially-observed data.

### E.4 MIWAE

**Missing data importance-weighted autoencoder (MIWAE)** employs deep generative models to perform missing data imputation (Mattei & Frellsen, 2019). Like MICE, it assumes that the data is MAR. The network is a variational autoencoder with an encoder and a decoder. We denote the latent variables by $Z$, their prior distribution by $p(Z)$ and the decoder by a function $p_\theta(X_{obs}|Z)$ parameterized by $\theta$. We further let $q(Z|X_{obs})$ be the variational distribution that approximates the true posterior $p(Z|X_{obs})$. The log-likelihood of the observed data distribution is then given by:

$$
\begin{aligned}
\log p(X_{obs}) &= \log \int p(X_{obs}|Z)p(Z)dZ \\
&= \log \int \frac{p(X_{obs}|Z)p(Z)}{q(Z|X_{obs})}q(Z|X_{obs})dZ \\
&= \log \mathbb{E}_{q(Z|X_{obs})}\left[\frac{p(X_{obs}|Z)p(Z)}{q(Z|X_{obs})}\right].
\end{aligned}
\tag{15}
$$

We can approximate the expectation inside the logarithm function of Eq.(15) using Monte Carlo sampling. Let $Z_i$ with $i \in \{1, \cdots, K\}$ be $K$ independent samples drawn from $q(Z|X_{obs})$, then Eq.(15) can be approximated as:

$$
\begin{aligned}
\log p(X_{obs}) &= \log \mathbb{E}_{q(Z|X_{obs})}\left[\frac{p(X_{obs}|Z)p(Z)}{q(Z|X_{obs})}\right] \\
&\approx \log \frac{1}{K}\sum_{i=1}^{K}\frac{p(X_{obs}|Z_i)p(Z_i)}{q(Z_i|X_{obs})}.
\end{aligned}
\tag{16}
$$

Optimizing the loss described in Eq.(16) leads to the design of the importance weighted autoencoder (IWAE). Once the network is trained, we can then use it to perform imputation. Let us consider the single imputation case, we aim to estimate the expectation of the missing variables' moments $h(X_{miss})$:

$$
\begin{aligned}
\mathbb{E}\left[h(X_{miss})|X_{obs}\right] &= \int h(X_{miss})p_\theta(X_{miss}|X_{obs})dX_{miss} \\
&= \int \int h(X_{miss})p_\theta(X_{miss}|X_{obs}, Z)p_\theta(Z|X_{obs})dZdX_{miss},
\end{aligned}
\tag{17}
$$

When $h$ is the identity function, $\mathbb{E}\left[h(X_{miss})|X_{obs}\right]$ becomes the conditional mean of the imputed missing variables given the observed variables. Since the ground-truth posterior distribution

Table 3: Hyperparameter range considered for grid search.

|  | Parameter | Grid |
|---|---|---|
| All models | Learning rate (LR) | [1.e-2, 5.e-3, 1.e-3] |
|  | Weight decay (WD) | [1.e-5, 1.e-4, 1.e-3] |
|  | MLP width | [50, 250, 500] |
|  | MLP depth | [1, 2, 5] |
| AE/MIWAE | Latent size | 25 |
|  | Encoder width | 128 |
|  | K | 20 |
| NeuMiss/NeuMISE | Number of blocks | 20 |

$p_\theta(Z|X_{obs})$ is intractable to compute, MIWAE learns an approximating posterior $q_\gamma(Z|X_{obs})$ using an encoder, from which the latent variables $Z$ are sampled. By employing the importance sampling technique (Tokdar & Kass, 2010), the conditional expected mean for missing variables can be approximated by:

$$\mathbb{E}\left[X_{miss}|X_{obs}\right] \approx \sum_{l=1}^{L} \omega_l X_{miss,l}, \qquad (18)$$

where $(X_{miss,1}, Z_1), \cdots, (X_{miss,L}, Z_L)$ are i.i.d. samples from $p_\theta(X_{miss}|X_{obs}, Z)q_\gamma(Z|X_{obs})$ via forward sampling. The weights $\omega_l$ are defined as:

$$\omega_l = \frac{r_l}{r_1 + \cdots + r_L} \quad \text{with} \quad r_l = \frac{p_\theta(X_{obs}|Z_l)p(Z_l)}{q_\gamma(Z_l|X_{obs})}, \qquad (19)$$

where $L$ is the number of decoding generated by sampling from the variational distribution $q_\gamma(Z|X_{obs})$. As a result, we fill in the missing values with their conditional mean. Moreover, multiple imputation approach can also be achieved using the importance sampling schemes. In this approach, denoting the number of multiple imputations by $M$ and assuming $L \gg M$, we resample $M$ times from the $L$ samples drawn from the neural network $(X_{miss,1}, Z_1), \cdots, (X_{miss,L}, Z_L)$.

### E.5 NEUMISS

Le Morvan et al. (2020) proposed NeuMiss as a novel neural network architecture to do supervised learning with missing data. NeuMiss was inspired by the analytical Bayes predictor for a linear model of multivariate normal covariates in Equation (12), which requires the calculation of the inverse covariance of the observed variables $\Sigma_{obs}^{-1}$. Rather than performing this inversion for every possible missingness pattern, NeuMiss approximates it via a differentiable Neumann series (Meyer, 2000, Chapter 3). A single Neumann iteration is implemented as a re-usable network block, where the non-linear activation is the multiplication by the inverted missingness indicator $\bar{m} = 1 - m$ (see Figure 1 and Le Morvan et al. (2020)). The unrolled version of the Neumann iterations are then implemented by concatenating these re-usable network blocks. Le Morvan et al. (2020) shows that NeuMiss scales well to high-dimensional features and is statistically efficient for medium-sized samples. Le Morvan et al. (2021) extends this result to non-linear outcome functions.

## F HYPERPARAMETER CHOICES

The range of hyperparameters considered in our experiments is detailed in Table 3. As mentioned in the main text, a grid search procedure was performed to identify the optimal parameters for a given setting. A separate set of parameters was chosen for each method (see experiment section for a list), dataset (high correlation, low correlation, LBIDD), environment (shift/no shift), and missing data mechanism (MCAR, non-monotone MAR, Gaussian self-masking, MAR-Y). The results of the hyperparameter search are listed in Table 4. The optimised hyperparameters mainly relate to the MLP predictor. Due to computational limitations and in line with previous work (Jarrett et al., 2022; Le Morvan et al., 2021), sensible defaults were chosen for AE/MIWAE and NeuMiss/NeuMISE. Optimising those may lead to some further peformance improvements for those methods.

Table 4: Final hyperparameters chosen via grid search, by experiment.

| Missing | Eval in | Corr | Method | LR | WD | MLP width | MLP depth |
|---|---|---|---|---|---|---|---|
| MCAR | No Shift | High | ICE+mask | 1.E-02 | 1.E-04 | 250 | 5 |
| | | | MICE | 1.E-02 | 1.E-04 | 250 | 5 |
| | | | AE+mask | 5.E-03 | 1.E-05 | 500 | 2 |
| | | | MIWAE | 1.E-03 | 1.E-05 | 50 | 2 |
| | | | NeuMiss | 1.E-03 | 1.E-04 | 250 | 2 |
| | | | NeuMISE | 1.E-03 | 1.E-04 | 500 | 5 |
| | | Low | ICE+mask | 5.E-03 | 1.E-03 | 250 | 5 |
| | | | MICE | 5.E-03 | 1.E-04 | 50 | 5 |
| | | | AE+mask | 5.E-03 | 1.E-05 | 50 | 2 |
| | | | MIWAE | 1.E-03 | 1.E-03 | 50 | 1 |
| | | | NeuMiss | 5.E-03 | 1.E-04 | 250 | 2 |
| | | | NeuMISE | 1.E-03 | 1.E-05 | 250 | 2 |
| | Shift | High | ICE+mask | 1.E-03 | 1.E-03 | 50 | 5 |
| | | | MICE | 5.E-03 | 1.E-04 | 250 | 5 |
| | | | AE+mask | 5.E-03 | 1.E-05 | 500 | 5 |
| | | | MIWAE | 1.E-03 | 1.E-03 | 50 | 5 |
| | | | NeuMiss | 1.E-03 | 1.E-03 | 500 | 2 |
| | | | NeuMISE | 1.E-03 | 1.E-05 | 500 | 5 |
| | | Low | ICE+mask | 5.E-03 | 1.E-05 | 50 | 1 |
| | | | MICE | 1.E-02 | 1.E-05 | 50 | 2 |
| | | | AE+mask | 5.E-03 | 1.E-05 | 250 | 1 |
| | | | MIWAE | 1.E-03 | 1.E-03 | 250 | 5 |
| | | | NeuMiss | 1.E-03 | 1.E-03 | 250 | 5 |
| | | | NeuMISE | 1.E-03 | 1.E-05 | 250 | 1 |
| Non-monotone MAR | No shift | High | ICE+mask | 5.E-03 | 1.E-03 | 500 | 5 |
| | | | MICE | 1.E-02 | 1.E-04 | 50 | 5 |
| | | | AE+mask | 1.E-02 | 1.E-04 | 50 | 5 |
| | | | MIWAE | 1.E-03 | 1.E-03 | 50 | 5 |
| | | | NeuMiss | 1.E-03 | 1.E-04 | 500 | 1 |
| | | | NeuMISE | 1.E-03 | 1.E-05 | 500 | 2 |
| | | Low | ICE+mask | 5.E-03 | 1.E-03 | 250 | 5 |
| | | | MICE | 5.E-03 | 1.E-04 | 50 | 5 |
| | | | AE+mask | 5.E-03 | 1.E-05 | 50 | 2 |
| | | | MIWAE | 1.E-03 | 1.E-03 | 250 | 2 |
| | | | NeuMiss | 1.E-03 | 1.E-04 | 250 | 5 |
| | | | NeuMISE | 1.E-03 | 1.E-05 | 500 | 5 |
| | Shift | High | ICE+mask | 1.E-03 | 1.E-03 | 50 | 5 |
| | | | MICE | 1.E-03 | 1.E-04 | 250 | 5 |
| | | | AE+mask | 5.E-03 | 1.E-05 | 500 | 5 |
| | | | MIWAE | 1.E-02 | 1.E-05 | 50 | 5 |
| | | | NeuMiss | 1.E-03 | 1.E-03 | 250 | 2 |
| | | | NeuMISE | 1.E-03 | 1.E-05 | 500 | 2 |
| | | Low | ICE+mask | 1.E-03 | 1.E-05 | 50 | 1 |
| | | | MICE | 5.E-03 | 1.E-05 | 50 | 2 |
| | | | AE+mask | 5.E-03 | 1.E-05 | 250 | 1 |
| | | | MIWAE | 1.E-03 | 1.E-03 | 500 | 5 |
| | | | NeuMiss | 1.E-03 | 1.E-03 | 250 | 5 |
| | | | NeuMISE | 1.E-03 | 1.E-05 | 250 | 5 |
| Gaussian SM | No shift | High | ICE+mask | 5.E-03 | 1.E-03 | 50 | 5 |
| | | | MICE | 5.E-03 | 1.E-04 | 500 | 5 |
| | | | AE+mask | 1.E-03 | 1.E-03 | 50 | 5 |
| | | | MIWAE | 1.E-03 | 1.E-03 | 250 | 5 |
| | | | NeuMiss | 1.E-03 | 1.E-03 | 500 | 1 |
| | | | NeuMISE | 1.E-03 | 1.E-05 | 250 | 5 |
| | | Low | ICE+mask | 5.E-03 | 1.E-03 | 50 | 5 |
| | | | MICE | 1.E-03 | 1.E-05 | 50 | 1 |

| Missing | Eval in | Corr | Method | LR | WD | MLP width | MLP depth |
|---|---|---|---|---|---|---|---|
| | | | AE+mask | 1.E-02 | 1.E-05 | 50 | 5 |
| | | | MIWAE | 1.E-03 | 1.E-03 | 500 | 5 |
| | | | NeuMiss | 1.E-03 | 1.E-04 | 250 | 5 |
| | | | NeuMISE | 1.E-03 | 1.E-05 | 250 | 2 |
| | Shift | High | ICE+mask | 1.E-03 | 1.E-03 | 250 | 5 |
| | | | MICE | 1.E-03 | 1.E-04 | 500 | 5 |
| | | | AE+mask | 1.E-03 | 1.E-03 | 50 | 5 |
| | | | MIWAE | 1.E-03 | 1.E-03 | 50 | 5 |
| | | | NeuMiss | 1.E-03 | 1.E-05 | 250 | 2 |
| | | | NeuMISE | 1.E-03 | 1.E-04 | 500 | 5 |
| | | Low | ICE+mask | 5.E-03 | 1.E-05 | 250 | 1 |
| | | | MICE | 5.E-03 | 1.E-05 | 50 | 5 |
| | | | AE+mask | 5.E-03 | 1.E-04 | 500 | 5 |
| | | | MIWAE | 1.E-03 | 1.E-03 | 250 | 2 |
| | | | NeuMiss | 1.E-03 | 1.E-05 | 250 | 5 |
| | | | NeuMISE | 1.E-03 | 1.E-05 | 250 | 2 |
| MAR-Y | No shift | High | ICE+mask | 1.E-03 | 1.E-04 | 500 | 5 |
| | | | MICE | 1.E-03 | 1.E-04 | 50 | 5 |
| | | | NeuMiss | 1.E-03 | 1.E-05 | 500 | 5 |
| | | | NeuMISE | 1.E-03 | 1.E-05 | 500 | 2 |
| | | | ICE+Y | 5.E-03 | 1.E-03 | 50 | 5 |
| | | | MICE+Y | 1.E-02 | 1.E-04 | 50 | 5 |
| | | Low | ICE+mask | 1.E-03 | 1.E-04 | 50 | 2 |
| | | | MICE | 1.E-03 | 1.E-04 | 50 | 2 |
| | | | NeuMiss | 1.E-03 | 1.E-05 | 500 | 5 |
| | | | NeuMISE | 1.E-03 | 1.E-05 | 500 | 2 |
| | | | ICE+Y | 1.E-02 | 1.E-05 | 50 | 5 |
| | | | MICE+Y | 5.E-03 | 1.E-03 | 250 | 5 |
| | Shift | High | ICE+mask | 5.E-03 | 1.E-05 | 50 | 5 |
| | | | MICE | 1.E-03 | 1.E-04 | 250 | 5 |
| | | | NeuMiss | 1.E-03 | 1.E-04 | 500 | 5 |
| | | | NeuMISE | 1.E-03 | 1.E-05 | 500 | 2 |
| | | | ICE+Y | 1.E-03 | 1.E-05 | 250 | 5 |
| | | | MICE+Y | 1.E-03 | 1.E-05 | 50 | 2 |
| | | Low | ICE+mask | 5.E-03 | 1.E-04 | 50 | 5 |
| | | | MICE | 5.E-03 | 1.E-05 | 50 | 5 |
| | | | NeuMiss | 1.E-03 | 1.E-05 | 500 | 2 |
| | | | NeuMISE | 1.E-03 | 1.E-05 | 500 | 1 |
| | | | ICE+Y | 1.E-02 | 1.E-03 | 500 | 5 |
| | | | MICE+Y | 1.E-03 | 1.E-04 | 500 | 1 |

