# OpenReview forum: "Robust prediction under missingness shifts"
_ICLR.cc/2024/Conference — Submitted to ICLR 2024_

### Official Review · Reviewer_ExJF · 2023-10-29

**Soundness:** 3 good
**Presentation:** 4 excellent
**Contribution:** 3 good
**Rating:** 6
**Confidence:** 5

**Summary:**

This paper examines theoretically and empirically the problem of missingness shift. It characterizes missingness shifts to which an optimal Bayes model would be robust, introduces an iterative imputation strategy for joint imputation and outcome modelling, and concludes with an empirical assessment of performance under missingness shifts.

**Strengths:**

This paper presents an excellent yet simple theoretical characterization of the missingness shift problem. The Bayes optimal formalisation is clearly described, and the connection with current approaches to missingness is appreciated. The paper is impeccably written, clear and well-motivated.

**Weaknesses:**

While the paper presents a strong and clear theoretical case, I believe addressing the following points would strengthen the paper, particularly concerning the empirical aspects:

- Existing works have studied the problem of missingness/observation shifts [1, 2]. One paper formalizes missingness shifts, while the other focuses on empirical studies. The authors should consider discussing the distinctions with their proposed formalization and approach.
- Equation (2) relies on the assumption: $\mathbb{E}[\epsilon \mid X_{obs}, M] = 0$. Authors should detail this assumption, its meaning and its real-world relevance. Particularly, does it not imply some independence of Y upon the missing data?
- In Section 4, there is an implicit assumption of no covariate or concept shift. Making this explicit would enhance clarity for readers.
- The paper would be strengthened by detailing NeuMISE and why this modelling would better approximate the conditional expectations and a Bayes optimal model.
- It would be beneficial to delve further into the discussion of Appendix F, particularly in addressing the unclear aspects of why the model performs less effectively when uncertainty increases.
- While the empirical results employ real-world covariates, the analysis lacks a study of real-world missingness shifts. Considering a dataset with changing missingness processes would strengthen the experimental analysis.
- Comparing the proposed approach with an end-to-end neural network that uses zero imputed data and mask as input seems a natural comparison to demonstrate the superiority of the proposed NeuMICE.


[1] Zhou, H., Balakrishnan, S., & Lipton, Z. (2023, April). Domain adaptation under missingness shift. In International Conference on Artificial Intelligence and Statistics (pp. 9577-9606). PMLR.
[2] Jeanselme, V., Martin, G., Peek, N., Sperrin, M., Tom, B., & Barrett, J. (2022). Deepjoint: Robust survival modelling under clinical presence shift. In NeurIPS Learning from Time Series for Health Workshop.

**Questions:**

I consider the paper to be a valuable theoretical contribution, my current rating is only hurt by the limitations outlined earlier.

---

> ### Author Response · Authors · 2023-11-22
> **Response to reviewer ExJF**
>
> We thank the reviewer for their time and constructive feedback! We would like to take the opportunity to address the points raised in the review. Since they were requested by multiple reviewers, a discussion of Zhou et al. (2023) as well as further details regarding NeuMISE are provided in a joint response above.
> \
> \
> \
> **Related work**
> For a detailed discussion of [1] Zhou et al. (2023), please see our joint response. We further added  [2] Jeanselme et al. (2022) to the related work section.
> \
> \
> \
> **Assumptions on the noise**
>
> The assumptions in Equation (2) are primarily needed for the formulation of the Bayes predictor in Equation (3), which under those assumptions simplifies to the expected value of a deterministic function of $X_{obs}$ and $X_{mis}$. However, all results still hold for weaker assumptions about the noise. We regret that this was not clear in our original submission and have revised the section accordingly.
>
> **Action:** We added additional text following Equation (2) as well as an intermediate step to Equation (3) to make the role of our assumptions clearer.
> \
> \
> \
> **Real-world data**
>
> We agree with the reviewer that real-world data that contains a missingness shift would further strengthen our results. Unfortunately, it is impossible to ascertain whether any observed shifts are ignorable or not. We therefore opted to focus on (semi-)simulated datasets with known missingness shifts, leaving the benchmarking of models on a wide range of real-world data for future work.
> \
> \
> \
> **Comparison to MLP with zero imputation**
>
> We included an MLP model with zero imputation in our early experiments. However, results from those early experiments showed such a model to be a weaker baseline than an MLP with ICE imputation. In the interest of manuscript space, a direct comparison of ICE and MICE, as well as already strained computational resources, we opted to omit zero imputation in favour of ICE. We believe that the most meaningful comparison of NeuMICE is to NeuMiss and ICE.
> \
> \
> \
> **Additional points**
>
> *Explicit assumption of no covariate/label shift:* We added an explicit statement of $p(Y, X) = q(Y, X)$ to the beginning of Section 4.
>
> *Extended discussion of results:* We further agree with the reviewer regarding Appendix F and have added some more interpretation to the results section, particularly related to the decreased performance of NeuMISE.
> \
> \
> \
> We believe that the above changes have strengthened the overall quality and contribution of the paper. We would like to kindly request that you reconsider the score assigned to our paper in light of the revisions. We believe the changes made have effectively addressed the issues you raised, and we hope this will be reflected in an updated evaluation.
>
> If you have any further questions or if there are specific areas you would like us to clarify, please do not hesitate to let us know.

---

> > ### Comment · Reviewer_ExJF · 2023-11-22
> > **Maintain score**
> >
> > Thank you very much for your answers and clarifications. Despite some of the results being previously known, I believe the paper remains an interesting contribution and I would like to maintain my score.

---

### Official Review · Reviewer_ntEa · 2023-10-30

**Soundness:** 3 good
**Presentation:** 2 fair
**Contribution:** 3 good
**Rating:** 8
**Confidence:** 3

**Summary:**

This work studies the problem of developing a robust predictive model in cases where the pattern through which covariates are missing changes between a source and target domain. The theoretical contributions of the work are a formalization of the problem, a proof that the optimal predictor is stable in case that the missingness pattern is ignorable in both environments (e.g. the missingness pattern depends only observed covariates), and an argument that predictors that leverage informative missingness can still be robust under missingness shift. They also introduce a new neural network architecture NeuMISE that builds off of the NeuMiss architecture for learning with missing data. Experiments with synthetic and real-world and data are conducted with injected missingess shift.

**Strengths:**

* This work tackles an important but under-emphasized problem with simple but powerful theoretical results. The core contribution regarding the formalization of missingness shift and discussion of ignorable shifts is a strong contribution and has potential for broad use in applications.
* The experiments are broad and cover a number of data generating processes, shift mechanisms, and comparator methods.

**Weaknesses:**

* The motivation for the NeuMISE method is not presented clearly enough or with enough detail to tie it to the rest of the core claims of the work. It is primarily not clear why modifying the masking of NeuMiss is well-motivated to address the issue of generalizing across unobserved missingness patterns.
* I have several concerns regarding the clarity of the work, which are elaborated on in the Questions section below.

**Questions:**

* Related to clarity:
  * Important aspects of the experiments are not presented clearly enough or with enough detail in the main text. For example, it is not explained what “low correlation” and “high correlation” corresponds to in the experiments.
  * Section 5 on the role of Y is interesting, but is not presented particularly clearly. In particular, please elaborate on how adjusting for Y in a source domain can induce missingness shift, but omitting Y results in a stable estimator.
  * The discussion focuses strongly on the comparing methods that leverage informative missingness vs. “unbiased” estimators that do not. This seems to be a critical point for the paper overall (e.g. related to the second of the three contributions listed in the introduction section), but it did not come through clearly to me in the writing. Furthermore, it is not clear how (or if) this was evaluated in the experiments. This could perhaps be improved with additional exposition earlier in the paper that sets up this argument more clearly with specific hypotheses to be evaluated.
* Can the results be generalized to binary outcomes? Naively, it seems like the additive noise model limits the direct applicability of this framework to binary outcomes.
* Is the assumption that the error term in equation (2) depends only on the observed X and M limiting the generalizability of this work? Which aspects of the results would no longer hold if the error term were to depend on the full X?
* Please comment on the relationship of the results of this work to Zhou et al 2023 “Domain Adaptation under Missingness Shift” and adjust claims regarding novelty and prior work, if appropriate.

---

> ### Author Response · Authors · 2023-11-22
> **Response to reviewer ntEa**
>
> We thank the reviewer for their time and constructive feedback. We would like to take the opportunity to address the points raised in the review. Since they were requested by multiple reviewers, a discussion of Zhou et al. (2023) as well as further details regarding NeuMISE are provided in a joint response above.
> \
> \
> \
> **Experimental detail in the main text**
>
> Due to the rather stringent space limitations, we limited the description of experiments to the minimum judged necessary to understand the results, providing the details required for full reproducibility in the appendix. We acknowledge that the final version may have allocated too little space to experimental description. We moved some of the experimental details back into the main text. In particular, we define exactly what we mean by “high” and “low” correlation in our experiments. We hope that we were able to strike the right balance between concision and detail.
>
> **Action:** We revised Section 7.1 to include additional details on the experimental setup.
> \
> \
> \
> **Clearer explanations of the role of Y**
>
> The Bayes predictor $\tilde f_m = E[Y \mid X_{obs}, M=m]$ is still well-defined for $Y$-dependent missingness. Furthermore, in the absence of a missingness shift, the Bayes predictor in source and target environment remain the same, i.e., $\tilde f_m = \tilde g_m$. Any method that successfully estimates $\tilde f_m$ without using $Y$ (i.e., omits $Y$) will therefore have unchanged performance in the target environment.
>
> However, the circumstances change, once we adjust for $Y$ by including it in an impute-then-regress model: we learn an estimator $\int \hat f (X_{mis}, X_{obs})p(X_{mis} \mid X_{obs}, Y)dX_{mis}$ that gives an unbiased estimate $\hat f$ of the complete data model $f^\star$ but requires knowledge about $Y$ — the outcome we want to predict. This is obviously problematic. To overcome this, others have proposed to learn $\hat f$ as above through the use of $p(X_{mis} \mid X_{obs}, Y)$ but then replace the imputation with $p(X_{mis} \mid X_{obs})$ in the target environment. We argue here that this does not estimate the Bayes predictor $\tilde g_m$.
>
> Please also see our reply to reviewer tBPA for some further explanations.
>
> **Action:** We revised the text in Section 6.1 (formerly 5.1) to clarify our argument.
> \
> \
> \
> **Informative missingness vs. “unbiased” estimation**
>
> One of the primary motivations for this research was indeed a common claim that unbiased estimation is preferable for clinical prediction modelling, usually implying that it leads to more robust models. However, we find that Theorem 1 does not provide an a priori reason why unbiased methods should be preferred for robust prediction. Instead, robust prediction under ignorable missingness shifts depends on the precise estimation of the Bayes predictors in the source environment, which may depend on both the method (e.g., impute-then-regress vs. end-to-end learning) and the data (e.g., levels of missingness).
>
> **Action:** We followed the advice of the reviewer and added a Section 4.1 that introduces the above argument. We further revised our section on “Learning under missingness shifts” to discuss certain conditions that may influence the estimation of the Bayes predictor.
> \
> \
> \
> **Additive noise and its independence of missing covariates**
>
> The reviewer is correct that neither the additive nature of the noise nor the assumption on its conditional mean are strictly necessary. We opted for this formulation, as it leads to a simple and clear expression of the optimal predictor given complete data in the form of $f^\star(X)$. However, our results remain valid for binary outcomes as well as non-additive noise that depends on the entire $X$.
>
> **Action:** We added additional text following Equation (2) in order to clarify the role of our assumptions. We also added Appendix A.2 to discuss the generality of our results.
> \
> \
> \
> We believe that the above changes have strengthened the overall quality and contribution of the paper. We would like to kindly request that you reconsider the score assigned to our paper in light of the revisions. We believe the changes made have effectively addressed the issues you raised, and we hope this will be reflected in an updated evaluation.
>
> If you have any further questions or if there are specific areas you would like us to clarify, please do not hesitate to let us know.

---

> > ### Comment · Reviewer_ntEa · 2023-11-22
> >
> > Thank you for the clarifications and revisions. I will update my score to 8.

---

### Official Review · Reviewer_P8Mn · 2023-10-31

**Soundness:** 3 good
**Presentation:** 3 good
**Contribution:** 2 fair
**Rating:** 3
**Confidence:** 4

**Summary:**

This work is motivated by the observation that the source of missingness (missing values in covariates) may differ in the train and deployment populations. The paper studies conditions under which the optimal predictor does not change in the presence of missingness shifts. It also analyzes the extent to which methods that utilize informative missingness generalize well in the presence of missingness shifts. They introduce a method called NeuMISE that aims to be robust across a range of missingness mechanisms.

**Strengths:**

1. Finding ways to cope with non-ignorable distribution shifts (shifts in the conditional distribution of Y|X) is an important and challenging problem and has not received as much attention as covariate distribution shift, so it’s great that this work points out that methods for dealing with missingness and ignorable missingness shift are insufficient in the presence of non-ignorable missingness shift.

2. The paper is clear and well-written.

**Weaknesses:**

1. The main weakness of this work is that it does not cite or discuss its connection to [1], which is another work that studies robustness to missingness shift. This paper describes that when missing data indicators are available, domain adaptation under missingness shift reduces to a covariate shift problem. This finding seems to be related to one of the central contributions of this paper, which is that the optimal predictor remains unchanged if missingness only depends on observables in both the training and test environment.


2. It’s not clear to me what advantage NeuMISE (the authors’ proposed method) has compared to the existing baseline in the presence of non-ignorable missingness shift. While the authors have some empirical results that NeuMISE performs outperforms other methods in the presence of non-ignorable missingness shift, I’m skeptical that such a result holds in general. To my understanding, generalizing well to non-ignorable missingness shift should only be possible if the model is in some sense robust to a variety of non-ignorable missingness shift, and I would presume that such a model may trade off some performance on the source data for better generalization across target environments. Is that the type of result that we observe for NeuMISE? What is the reason that NeuMISE is more robust? Furthermore, what benefit does NeuMISE offer compared to existing baselines.

3. It would be helpful to add a few concrete examples where missingness shifts occur in the real-world to motivate the research.

Improvements:

1. It would be helpful if the authors emphasize in their abstract/introduction that they focus on missingness in covariates, not labels. There is an extensive literature on learning with missing labels and it is somewhat unclear what type of missingness the authors are focusing on until the problem definition in Section 3.

2. It would be nice to draw a connection between ignorable / non-ignorable missingness to ignorable / non-ignorable sample selection.

[1] Zhou, Helen, Sivaraman Balakrishnan, and Zachary Lipton. "Domain adaptation under missingness shift." International Conference on Artificial Intelligence and Statistics. PMLR, 2023.

**Questions:**

1. It would be helpful if the authors add a line after Equation 2 that explains what the assumption $\mathbb{E}[ \epsilon \mid X_{obs}, M] = 0$ means concretely – i.e., to what extent can the noise $\epsilon$ depend on the missingness $M$? The current presentation does not require $\epsilon$ to be independent of $M$ – is that the desired interpretation? To the best of my understanding, in the current presentation, the variance of the noise $\epsilon$ could depend on the missingness mechanism.

2. Could the authors explain why the following is true: ``If $Y$ only influences missingness in the source environment, shifts may still be ignorable.”

---

> ### Author Response · Authors · 2023-11-22
> **Response to reviewer P8Mn**
>
> We thank the reviewer for their time and constructive feedback. We would like to take the opportunity to address the points raised in the review. Since they were requested by multiple reviewers, a discussion of Zhou et al. (2023) as well as further details regarding NeuMISE are provided in a joint response above.
> \
> \
> \
> **Assumptions about noise**
>
> In our formulation, the noise may indeed depend on the missingness. That is, missingness may actively (and even causally) influence Y. Consider for example a situation in which a biomarker A is used by doctors to decide whether another test B should be performed, which requires referral to a specialist laboratory. If the patient is referred, the laboratory also measures Y. If the patient isn’t referred, the doctor themselves measures Y. If the laboratory has a more precise machine than the doctor then $Var(\epsilon \mid A, M_B=0) < Var(\epsilon \mid A, M_B=1)$ but $E(\epsilon \mid A, M) = 0$ may still be true. Our theory would allow for such a scenario.
>
> We would further like to point out that the assumptions in Equation (2) are primarily needed for the formulation of the Bayes predictor in Equation (3), which under those assumptions simplifies to the expected value of a deterministic function of $X_{obs}$ and $X_{mis}$. Please also refer to our reply to reviewer ntEa regarding the possibility of non-additive noise or noise that depends on $X_{mis}$.
>
> **Action:** We added additional text following Equation (2) as well as an intermediate step to Equation (3) to clarify the role of our assumptions.
> \
> \
> \
> **Ignorability of Y-dependent missingness in the source environment**
>
> Since Y is observed at training time, Y-dependent missingness in the source environment can be MAR if the imputation model conditions on $Y$. The source missingness is thus ignorable. If a shift occurs through which the target missingness no longer depends on Y and only depends $X_{obs}$, then the target missingness is also ignorable. By Definition 2, the missingness shift is therefore ignorable.
> The main difficulty then lies in how to derive an $\tilde{f}^\star_m$ that does not require $Y$ in order to be used in the target environment (since we will not have $Y$ when we want to predict). Mathematically speaking, we need to find a way to derive $q(X_{mis} \mid X_{obs})$ from $p(X_{mis} \mid X_{obs}, Y)$. We added a sketch of how this might be achieved to the appendix, although we would like to stress that we haven’t been able to verify it empirically.
>
> **Action:** We revised Section 6.2 (formerly 5.2) of our paper to make this argument clearer. We further outlined a simple two-step impute-then-regress approach in the Appendix that may be used to derive a Bayes estimator for the target environment from the source data.
> \
> \
> \
> **Additional points**
>
> *Real-world examples:* Our original introduction contained two examples of real-world missingness shifts: 1) a reduction in a test’s costs may increase its use, making it more readily available for a larger share of patients and 2) the deployment of the model in clinical practice may change how doctors collect data, for example by more commonly collecting the variables that are used by the model. We’ve added a third example: a change in clinical guidelines that govern how and when doctors ought to perform clinical examinations. Each of these cases changes the missingness in the data and represents an example of real-world missingness shift. All examples focus on healthcare, as this is our main domain of expertise. We believe that the importance of medical applications — and the role that AI can play within it — serve as sufficient motivation but we are certain similar examples could be found for other domains.
>
> *Focus on covariate shift:* We added a clarification to the abstract and introduction, highlighting that our work focuses on missingness in covariates.
> \
> \
> \
> We believe that the above changes have strengthened the overall quality and contribution of the paper. We would like to kindly request that you reconsider the score assigned to our paper in light of the revisions. We believe the changes made have effectively addressed the issues you raised, and we hope this will be reflected in an updated evaluation.
>
> If you have any further questions or if there are specific areas you would like us to clarify, please do not hesitate to let us know.

---

> > ### Author Response · Authors · 2023-11-23
> >
> > Dear reviewer,
> >
> > As the discussion period is drawing to a close in a few hours, we would like to thank you once again for your thoughtful consideration. We greatly appreciate the time and effort you have already dedicated to evaluating our work.
> >
> > If you could spare a moment to revisit our rebuttal, we would be immensely grateful. We carefully addressed your suggestions and we believe that these adjustments have significantly strengthened the overall quality of our paper. In particular, we **clarified the connection of our work to Zhou et al. (2023)** and **improved our discussion of NeuMISE**.
> >
> > In light of the above changes, we would greatly appreciate your feedback so your reviews and score may reflect the updated material.
> >
> > Kind regards,
> >
> > The authors

---

### Official Review · Reviewer_tBPA · 2023-11-02

**Soundness:** 2 fair
**Presentation:** 3 good
**Contribution:** 2 fair
**Rating:** 5
**Confidence:** 3

**Summary:**

This paper considers a prediction problem with missing shift settings, which is an important practical task.  The paper first provides an overall review of missing mechnisms and related literature. It further discusses the equivalence of Bayes predictors under ignorable missing shift and the effects of shifts in Y-dependence. It also proposes a NeuMISE to handle this challenging task.

**Strengths:**

The paper is well-written and has a very clear organization. The proposed method, NeuMISE, seems to be simple but effective and outperform other baselines. The results are relatively complete and solid.

**Weaknesses:**

The paper uses quite a lot space to discuss the missingness shift. Although such descriptions are complete and clear, it seems to be relatively elementary and do not provide enough new intelletucal insights. Under ignorable condition, Theorem 1 "equivalence" is also straightforward and hence is not surprising, at least to me.

Last few sentences in Section 5.1 confuse me. what is "adjusting Y", "omitting Y" and definition of "stable estimator"?

Section 6 is rather short. It should be expanded to explain why NeuMISE is more effective from a deeper viewpoint.

**Questions:**

See weakness points.

**Details Of Ethics Concerns:**

No Ethic Concerns.

---

> ### Author Response · Authors · 2023-11-22
> **Response to reviewer tBPA**
>
> We thank the reviewer for their time and constructive feedback. We would like to take the opportunity to address the points raised in the review. Since it was requested by multiple reviewers, further details regarding NeuMISE are provided in a joint response above.
> \
> \
> \
> **Too much detail in descriptions**
>
> We aimed for maximum clarity in our description of the problem setting and background. We appreciate the reviewer’s feedback and tried to remove some of the detail in favour of other, more insightful, sections.
>
> **Action:** We removed some of the detail in Section 3. However, due to additional content requested by other reviewers, the section is only marginally shorter in the revised manuscript.
> \
> \
> \
> **Adjusting/omitting Y**
>
> In Section 5.1, we used “adjusting Y” to refer to the inclusion of the outcome in the imputation model $p(X_{mis} \mid X_{obs}, Y)$. This is standard practice in statistical inference under Y-dependent missingness, as it results in a MAR setting and allows for unbiased estimation. In contrast, by “omitting Y” we mean an imputation model $p(X_{mis} \mid X_{obs})$ that does not include $Y$ in the conditioning set.
>
> We consider an estimator to be stable if it works equally well in the source and target environment. In the absence of a missingness change, $\tilde f_m = \tilde g_m$ is trivially true and any estimator that applies the function it learned from the source distribution to the target distribution will be stable. However, a method that tries to adjust for $Y$ during training will learn $\int \hat{f}(X_{mis}, X_{obs})p(X_{mis} \mid X_{obs}, Y)dX_{mis}$ but — since $Y$ cannot be observed in the target environment — would need to apply $\int \hat{f}(X_{mis}, X_{obs})q(X_{mis} \mid X_{obs})dX_{mis}$ at deployment. These two functions differ in general. The unobservable nature of $Y$ therefore introduces an “artificial” missingness shift, rendering a setting that is MAR at training time MNAR at deployment time.
>
> **Action:** We revised the text in Section 6.1 (formerly 5.1) to clarify our argument.
> \
> \
> \
> We believe that the above changes have strengthened the overall quality and contribution of the paper. We would like to kindly request that you reconsider the score assigned to our paper in light of the revisions. We believe the changes made have effectively addressed the issues you raised, and we hope this will be reflected in an updated evaluation.
>
> If you have any further questions or if there are specific areas you would like us to clarify, please do not hesitate to let us know.

---

> > ### Author Response · Authors · 2023-11-23
> >
> > Dear reviewer,
> >
> > As the discussion period is drawing to a close in a few hours, we would like to thank you once again for your thoughtful consideration. We greatly appreciate the time and effort you have already dedicated to evaluating our work.
> >
> > If you could spare a moment to revisit our rebuttal, we would be immensely grateful. We carefully addressed your suggestions and we believe that these adjustments have significantly strengthened the overall quality of our paper. In particular, we **expanded our discussion of NeuMISE**, discussing its motivation and principles in more depth.
> >
> > In light of the above changes, we would greatly appreciate your feedback so your reviews and score may reflect the updated material.
> >
> > Kind regards,
> >
> > The authors

---

### Author Response · Authors · 2023-11-21
**Joint response to the reviewer's comments**

Dear reviewers,

We thank you very much for your time and constructive feedback! Since several reviewers raised similar points, we decided to address them in a joint response here. All remaining questions are addressed in individual responses to each reviewer’s comments.


**Related work (Reviewers: P8Mn, ntEa, ExJF)**

We thank the reviewers for bringing Zhou et al. (2023) to our attention. We regret that we were not aware of this recent work.

Proposition 1 and its proof in Appendix B of Zhou et al. are indeed closely related to our Theorem 1. In it, Zhou et al. show that under MCAR and (v-)MAR, the conditional distribution $p(Y | X_{obs}, M)$ satisfies the covariate shift assumption. This is another view on our result that the Bayes predictor remains unchanged under ignorable missingness shifts. However, Zhou et al. stop their investigation there, with the rest of the work focusing on domain adaptation in situations where it is unclear which entries were actually missing, which differs from the question we are trying to answer in our work.

Therefore, although Zhou et al. share important findings with our work, we believe that our study adds several novel and important contributions:
- **We investigate missingness shift through a missing data lens:** Zhou et al. approach the problem from a domain adaptation perspective. While this is valid, it is an unusual view of missing data problems. We complement this view by tying Theorem 1 to the rich literature on missing data. Through this connection, one can leverage the well-established missing data theory on the results of our analysis.
- **We discuss the implications of Theorem 1 for the most common missing data approaches:** Zhou et al. stop their investigations after deriving Proposition 1. They do not consider its impact on the traditional imputation models (e.g., MICE) or recent advances in predicting with missing data (e.g., NeuMiss). In our work, we look at these methods in detail. In particular, we show that either may lead to robust prediction under ignorable shifts, thus questioning the imperative of unbiased estimation for robust prediction.
- **We relate domain adaptation methods to traditional missing data methods:** Zhou et al. only briefly mention that classical domain adaptation methods such as re-weighting with $q(x)/p(x)$ may be used in the case of shared support between source and target environments. We added a comment to the revised manuscript on how this may be viewed as a classical method for dealing with missing data: inverse probability weighting. We would also like to point out that imputation can similarly be seen as a domain adaptation method: by creating a fully observed dataset $\mathcal{D}^*$, imputation removes any covariate shift caused by missingness shifts— recasting the standard practice of data imputation as an effective solution to covariate shift due to changes in missingness.
- **We investigate the role of Y:** Zhou et al. limit their investigations to missingness that only depends on $X_{obs}$. We extend this to missingness that may depend on Y. In particular, we show that Y-dependent missingness shifts will almost always be non-ignorable. We further show that the common practice of using $Y$ for imputation in the imputation model will lead to an “artificial” missingness shift.
- **We perform extensive empirical evaluations:** Zhou et al. did not perform any empirical investigation of their Proposition. We empirically investigate the impact of different types of missingness shift on various estimators. By doing so, we are able to show that different estimators are more or less robust under different shifts.

**Action:** We added Zhou et al. (2023) to the related work and discussed it in our derivation of Theorem 1. We further adjusted our contributions and claims about novelty accordingly.

---

> ### Author Response · Authors · 2023-11-22
> **Continued joint response**
>
> **Motivation for NeuMISE**
>
> The reviewers have requested further detail on NeuMISE and why it might be more robust to missingness shifts. NeuMISE was inspired by the observation that pattern submodels like NeuMiss have greater difficulties extrapolating to previously unseen missingness patterns, especially if they are far away from the training data. In particular, when considering fully observed data, NeuMiss would still embed the fully observed vector $X$ as $\Sigma^{-1} X$, where $\Sigma^{-1}$ is the inverted observed-data covariance. In the case of ignorable missingness, however, it would be sufficient to forward the unchanged $X$. NeuMiss thus adds unnecessary noise to the input. By modifying the mask, we change this behaviour: rather than embedding with $\Sigma_{obs}^{-1}X_{obs}$ like NeuMiss, we impute with $\Sigma_{mis,obs}\Sigma_{obs}^{-1}X_{obs}$. Importantly, the observed $X_{obs}$ remain unchanged and are forwarded to the prediction layers as is.
>
> We further would like to point out that we do not intend to claim that NeuMISE remains robust across all possible missingness shifts, including those that are non-ignorable. Reviewer P8Mn is perfectly right in stating that “generalizing well to non-ignorable missingness shift should only be possible if the model is in some sense robust to a variety of non-ignorable missingness shift and I would presume that such a model may trade off some performance on the source data for better generalization across target environments”. In fact, our experiments on Y-dependent missingness clearly show this, and one would be much better off using MICE+Y. This is directly related to the trade-off mentioned by reviewer P8Mn, as MICE+Y performs much worse in the source environment. However, we observed empirically that NeuMISE consistently performed more robustly or on par with NeuMiss in both shifted and unshifted environments. We have revised some of the text to make this clearer and avoid false claims of robustness across any missingness shift.
>
> **Action:** We revised section 5 (formerly section 6), and in particular section 5.1 (formerly section 6.2) to provide further detail on NeuMISE and why modifying the mask may be well-motivated.
>
>
> **Changelog:**
> - We revised our abstract and introduction to adjust our claims regarding novelty. We  also clarified our focus on missingness in covariates and added another real-world example of missingness shifts.
> - We added Zhou et al. (2023) and Jeanselme et al. (2022) to the related work in Section 2.
> - We explain the impact of our assumptions on the noise in Section 3.
> - We updated our description of Theorem 1 in Section 4, prominently referring to Zhou et al.
> - We added a discussion on the importance of unbiased estimation for robust prediction under missingness shifts to Section 4.
> - We moved former Section 4.2 “Note on changes in Bayes risk” to the Appendix in order to save space.
> - We switched former Sections 5 and 6 and revised them in response to the reviewers’ comments.
> - We slightly extended the discussion of the empirical results.
> - We changed the description of the missingness mechanism from Monotone MAR to Non-monotone MAR, as this was falsely labelled in the original submission.
> - We added further explanations and details to the appendix.
>
>
> We believe that the above changes have strengthened the overall quality and contribution of the paper. We would like to kindly request that you reconsider the score assigned to our paper in light of the revisions. We believe the changes made have effectively addressed the issues you raised, and we hope this will be reflected in an updated evaluation.
>
> If you have any further questions or if there are specific areas you would like us to clarify, please do not hesitate to let us know and we will still try to provide an answer before the deadline.

---

### Meta-Review · Area_Chair_oMwt · 2023-12-11

**Metareview:**

This is an interesting work on the consequences of a shift in the missingness process. This might be considered a niche topic for the conference, which is nevertheless important. Some motivation issues have been raised. The experimental analysis can be considered as limited for this type of submission, and it is a point for improvement. The relation to Zhou et al. has led to a concern about the amount of novelty.

**Justification For Why Not Higher Score:**

Motivation, comparison and novelty wrt previous work, and experimental analysis could have been better presented.

**Justification For Why Not Lower Score:**

N/A

---

### Decision · Program_Chairs · 2024-01-16

Reject